# RL-DARTS: DIFFERENTIABLE ARCHITECTURE SEARCH FOR REINFORCEMENT LEARNING

## ABSTRACT

Recently, Differentiable Architecture Search (DARTS) has become one of the most popular Neural Architecture Search (NAS) methods successfully applied in supervised learning (SL). However, its applications in other domains, in particular for reinforcement learning (RL), has seldom been studied. This is due in part to RL possessing a significantly different optimization paradigm than SL, especially with regards to the notion of replay data, which is continually generated via inference in RL. In this paper, we introduce RL-DARTS, one of the first applications of end-to-end DARTS in RL to search for convolutional cells, applied to the challenging, infinitely procedurally generated Procgen benchmark. We demonstrate that the benefits of DARTS become amplified when applied to RL, namely search efficiency in terms of time and compute, as well as simplicity in integration with complex preexisting RL code via simply replacing the image encoder with a DARTS supernet, compatible with both off-policy and on-policy RL algorithms. At the same time however, we provide one of the first extensive studies of DARTS outside of the standard fixed dataset setting in SL via RL-DARTS. We show that throughout training, the supernet gradually learns better cells, leading to alternative architectures which can be highly competitive against manually designed policies, but also verify previous design choices for RL policies.

## 1 INTRODUCTION AND MOTIVATION

Over the last decade, recent advancements in deep reinforcement learning have heavily focused on algorithmic improvements (Hessel et al., 2018; Wang et al., 2016), data augmentation (Raileanu et al., 2020; Kostrikov et al., 2020), infrastructure upgrades (Espeholt et al., 2018; Horgan et al., 2018), and even hyperparameter tuning (Zhang et al., 2021; Faust et al., 2019). The automation of such RL methods have given rise to the field of Automated Reinforcement Learning (AutoRL). Surprisingly however, one relatively underdeveloped and unexplored area is automating large-scale policy architecture design.

Recently, works in larger-scale RL suggest that designing a policy's architecture can be just as important as the algorithm or quality of data, if not more, for various metrics such as generalization, transferrability, and efficiency. One surprising phenomenon found on Procgen (Cobbe et al., 2020), a procedural generation benchmark for RL, was that "IMPALA-CNN", a residual convolutional architecture from (Espeholt et al., 2018), could substantially outperform "NatureCNN", the standard 3-layer convolutional architecture used for Atari (Mnih et al., 2013), in both generalization and sample complexity under limited and infinite data regimes respectively (Cobbe et al., 2019). Furthermore, in robotics subfields such as grasping (Kalashnikov et al., 2018; Rao et al., 2020), cameras collect very detailed real-world images (ex: 472 x 472, 4x larger than ImageNet (Russakovsky et al., 2015)) for observations which require deep image encoder architectures, consisting of more than 15 convolutions, raising concerns on efficiency and speed in policy training and inference. As such RL policy networks gradually become larger and more sophisticated, so does the need for understanding and automating such designs.

Many efficient Neural Architecture Search (NAS) methods utilize the *one-shot* setup, where only one set of shared weight parameters is trained through an ensemble of models, otherwise known as a *supernet*. Differentiable Architecture Search (Liu et al., 2019), trains the supernet by inserting architecture selection variables inside the network in a continuous relaxation approach, in order to be differentiable with respect to a loss. These architecture search parameters are then used to select a final discretized perception module to be retrained from scratch in an evaluation phase.

The differentiable aspect of DARTS is highly attractive to RL, which possess two complications which SL does not:

1. RL is an *entire system*. In most RL algorithms involving auto-differentiation, there are multiple different phases: a *collection/exploration phase* which collects environment data via forward passes/inference of a policy, in addition to a *training phase* which updates parameters for potentially multiple network components (e.g. both the policy and value function in policy gradient methods). There can also even be a separate *evaluation phase* (disjoint from evaluation in NAS) different from the collection phase (e.g. using argmax greedy evaluation instead of the stochastic actor, for both Q-learning and policy gradient methods). Such phases are also commonly distributed over different machines, adding an extra layer of complexity via inter-machine communication.

2. RL training curves **possess considerably more variance and noise** than SL curves, with a much larger room for error due to sensitivity to hyperparameters (Andrychowicz et al., 2020), which originally led to the criterion of reporting mean and standard deviation for 3 seeded runs (Henderson et al., 2018). In fact, this noise can be nearly *unbounded*, as in many cases, an unlucky and catastrophic seeded training run can simply lead to a trivial reward even with an optimal architecture. Such noise may potentially be problematic for fast convergence for blackbox/evolutionary optimization methods. Furthermore, the RL **loss contains important information not conveyed by just pure reward** (e.g. value function error and entropy penalty, as well as potentially exploration/novelty bonuses), whereas information may be lost when simply summarizing all progress to a single scalar point of data.

DARTS solves (1) as it integrates naturally within the RL system with **minimal modifications** to the code and pipeline, by simply replacing the image encoder in standard RL algorithms. For (2), architecture search parameters in the DARTS supernet are joint trained alongside weight parameters, and can thus also use information from multiple auxiliary losses, which can be much richer signals than potentially sparse rewards. We represent this approach in Figure 1.

However, since there has been no previous proof of DARTS's feasibility in standard RL, one may hypothesize a few subtle concerns. While DARTS in SL is simpler and can focus more on generalization, a core major challenge in RL-DARTS is the feasability of training the supernet itself due to a non-stationary data distribution. This is especially because using a differentiable loss (ex: Bellman error) as the search objective is a double-edged sword in RL, as loss is defined with **respect to the replay buffer, whose data is collected by the policy itself via inference on the architecture.** In SL, the data distribution is independent of the architecture, and the loss (e.g. cross-entropy or negative log-likelihood)

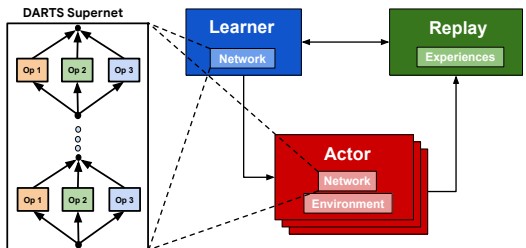

**Figure 1:** Representation of our method, in which a DARTS supernet is simply inserted into the network components of a standard RL training pipeline, which may potentially be highly distributed.

is strongly correlated or even equivalent to the objective (e.g. accuracy or density estimation (Parmar et al., 2018)). However **in RL, a bad architecture could lead to a poor initial distribution of data, and thus a local optimum where the loss is zero while the policy is still poor.** This is inherently due to the fact that the data in the replay buffer may not sufficiently cover the state and action space of the entire environment, and is common in cases such as sparse reward problems. This issue occurs even if the replay buffer data is fixed, which is the basis of offline RL (Levine et al., 2020).

The other question occurs during the *discretization* phase of DARTS, as any continuous relaxation method must deal with *integrality gaps* when rounding back to a discrete space, with this gap dependent on the optimization landscape. At the core of multiple discretization procedures (Liu et al., 2019; Chu et al., 2020c; Hundt et al., 2019) lies the idea of op selection based on edge weights. However, RL's optimization landscape is significantly different (Ilyas et al., 2020; Ahmed et al., 2019) from SL, and thus **even if the supernet trains in RL, it is unclear if e.g. DARTS's default discretization procedure leads to a better discrete cell, which is the original goal of DARTS and NAS in general.**

In this paper, we carefully analyze the extent at which DARTS can optimize perception modules for RL environments. While there is a rich series of active study on the theoretical and empirical behavior of DARTS optimization in SL (Wang et al., 2021b; Chen & Hsieh, 2020; Li et al., 2021), **our work is one of the first to study its optimization in RL, a completely different paradigm. Through extensive experiments, we ultimately validate that even vanilla DARTS is capable of architecture search in RL, laying a core foundation for future work in NAS for RL.** In summary, our contributions are:

- We demonstrate DARTS can be integrated with existing RL algorithms in a minimally invasive and easy-to-use manner, agnostic to the specific pipeline. Remarkably, on the challenging Procgen benchmark featuring infinite data, the DARTS supernet is able to train from scratch and efficiently produce similar training curves as baselines using both on-policy (PPO) and off-policy (Rainbow) methods, both of which require inference on the network.

- Due to the common issue of integrality gaps between supernet cells and discrete cells, we verify both qualitatively via visualizations and quantitatively via rewards, that discretized cells start with suboptimal architectures and gradually evolve to better architectures. However, we also demonstrate how this can fail, especially if the corresponding supernet fails to train, with further extensive ablations in the Appendix.

- By evaluating the final discrete cells at the end of supernet training, we discover that many environments benefit from learned, custom-tailored architectures, suggesting that human-designed architectures can be suboptimal. Furthermore, we find that joint-training a supernet on multiple environments can potentially produce a single transferrable cell. This opens up questions for future study, as large-scale RL can potentially benefit from more applications of NAS techniques.

## 2 RELATED WORKS

Most previous AutoRL methods can be thought of in terms of blackbox and evolutionary methods. For NAS on RL, these include (Song et al., 2021; Gaier & Ha, 2019; Stanley & Miikkulainen, 2002; Stanley et al., 2009) which utilize CPU workers for forward pass evaluations rather than exact gradient computation on GPUs. Such methods are usually unable to train policies involving more than 10K+ parameters due to the sample complexity of zeroth order methods in high dimensional parameter space (Agarwal et al., 2010). However, such methods have proved useful on smaller search spaces for RL, such as hyperparameter optimization (Zhang et al., 2021; Franke et al., 2020; Parker-Holder et al., 2020; Jaderberg et al., 2017) and algorithm search/"learning to learn" methods (Faust et al., 2019; Co-Reyes et al., 2021).

In terms of differentiable search in RL, methods for tuning hyperparameters and environments include (Zahavy et al., 2020; Xu et al., 2018; Hu et al., 2020). The only previous known application of DARTS is (Akinola et al., 2021), which searches for the optimal way of combining the observation and action tensors together in off-policy QT-OPT (Kalashnikov et al., 2018). However, this approach does not modify the image encoder nor uses the supernet for inference, as off-policy robotic data collected independently from the actual policy is used for training, which leaves the applicability of DARTS to inference-dependent RL as an open question addressed in our work.

## 3 METHODOLOGY

### 3.1 VANILLA DARTS AND NAS PRELIMINARIES

We provide a brief review of DARTS which can be pictorially summarized in Figure 2, although more details can be found in the original DARTS paper (Liu et al., 2019). DARTS optimizes substructures called cells, where each cell contains $I$ intermediate nodes organized in a directed acyclic graph, where each node $x^{(i)}$, represents a feature map, and each edge $(i, j)$ consists of an operation (op) $o^{(i,j)}$, with later nodes $x^{(j)}$ merged (e.g. summation) from some previous $o^{(i,j)}(x^{(i)})$. A DARTS supernet is constructed by continuously relaxing selection of ops in $\mathcal{O}$, via softmax weighting, i.e. $\overline{o}^{(i,j)}(x^{(i)}) = \sum_{o \in \mathcal{O}} p_o^{(i,j)} \cdot o(x^{(i)})$, where $p_o^{(i,j)} = \frac{\exp(\alpha_o^{(i,j)})}{\sum_{o' \in \mathcal{O}} \exp(\alpha_{o'}^{(i,j)})}$. The cell's output is by default

the result of a Conv1x1 op on the depthwise concatenation of all intermediate node features, although this may be changed (e.g. simply output last intermediate node's features). We denote the collection of all architecture variables $a_o^{(i,j)}$ as $\alpha$. Since the total op space $\mathcal{O}$ must contain the Zero and Skip Connection ops, we mainly refer to the main op space as $\mathcal{O}_{base}$ where $\mathcal{O} = \mathcal{O}_{base} \cup \{Zero, Skip\}$. In the standard optimization framework for SL where there is the notion of a validation set, the learning procedure consists of a bilevel optimization problem:

$$\alpha^* = \arg\min_\alpha \mathcal{L}_{val}(\theta^*, \alpha) \quad \text{s.t.} \quad \theta^* = \arg\min_\theta \mathcal{L}_{train}(\theta, \alpha) \tag{1}$$

where $\mathcal{L}(\theta, \alpha)$ is a loss function defined with respect to the supernet. $\alpha^*$ is then *discretized* into a final cell, by representing each edge $(i, j)$ with the highest softmax weighted op $\arg\max_{o \in \mathcal{O}, o \neq zero} p_o^{(i,j)}$ and then retaining only the top $K = 2$ incoming edges for each intermediate node.

Following common NAS practices (Zoph et al., 2018; Pham et al., 2018; Zoph & Le, 2016), we construct our supernet by stacking both normal ($N$ times) and reduction cells ($R$ times) together into *blocks*, which are themselves also stacked together $D$ times (see Figure 2). Normal cells preserve the input shape on output, while reduction cells apply a stride of 2 to reduce height and width immediately on the input. Each block can have its own associated convolutional channel depth which is used throughout all cells in the block. Another common NAS practice we adopt is to train a smaller supernet (i.e. depth 16) to reduce computation time, but evaluate final discretized cells on larger models (i.e. depth 64), with $D = 3$ layers for cheap large-scale runs and $D = 5$ layers for fine-grained A/B testing.

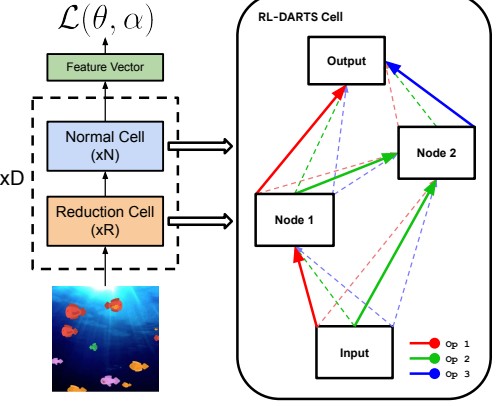

**Figure 2:** Illustration of our supernet via stacking normal and reduction cells. Solid lines correspond to selected ops after discretizing the supernet, which contains all possible ops weighted using $\alpha$. If $R > 0$, we add an inital Conv3x3 for preprocessing.

### 3.2   RL-DARTS

For notation, we use standard conventions in the RL literature: for a given MDP $\mathcal{M}$, we denote $s_t, a_t, r_t$ as the state, action, reward respectively at time $t$. $\pi$ is the policy and $\mathcal{D}$ is the replay buffer which contains collected trajectories $\tau = (s_0, a_0, r_0, s_1, \ldots)$. The goal of the RL algorithm is to maximize $J(\pi) = \mathbb{E}_{\tau \sim \pi}\left[\sum_{t \geq 0} r_t\right]$, the expected cumulative reward when using policy $\pi$. In most RL algorithms, there is the notion of a neural network torso, or *encoder* which maps a state $s$ (or sequence $\tau$) to a feature vector, which may be projected to correct action dimensions, behind the scenes in forming $\pi$. In the DARTS case, we use a supernet for this encoder instead, which leads to a policy denoted as $\pi_{\theta, \alpha}$.

The original bilevel optimization framework (Eq. 1) is notoriously difficult and unstable, sometimes requiring special techniques (Liu et al., 2018; Dong & Yang, 2019; Hundt et al., 2019; Li et al., 2020; Chu et al., 2020b;a;c; Liang et al., 2019; Wang et al., 2021b) specific to SL optimization. To avoid such confounding factors, especially when DARTS has not yet been fully

---

**Algorithm 1: RL-DARTS Procedure.**

**1. Supernet training:** Compute $\alpha^*$ from $\arg\max_{\theta,\alpha} J(\pi_{\theta,\alpha})$.

**2. Discretization:** Discretize $\alpha^*$ to construct evaluation policy $\pi_{\phi,\delta(\alpha^*)}$.

**3. Evaluation:** Report $\max_\phi J(\pi_{\phi,\delta(\alpha^*)})$.

---

validated in RL, we may simply optimize sample complexity and raw training performance which are common metrics in RL. We thus joint optimize both $\theta$ and $\alpha$:

$$\arg\max_{\theta,\alpha} J(\pi_{\theta,\alpha}) \quad \text{via} \quad \arg\min_{\theta,\alpha} \mathcal{L}(\theta, \alpha) \tag{2}$$

which leads to an optimal $\alpha^*$. With slight abuse of notation for convenience, denote $\pi_{\phi,\delta(\alpha^*)}$ as the policy using the discretized cell $\delta(\alpha^*)$ and $\mathcal{L}_{\delta(\alpha^*)}(\phi)$ to be the corresponding loss function, now dependent on only the new sparse weights $\phi$. Note that under standard NAS procedure, we must then *evaluate* the discretized cell via training from scratch again to obtain the final reward:

$$\max_\phi J(\pi_{\phi,\delta(\alpha^*)}) \quad \text{via} \quad \arg\min_\phi \mathcal{L}_{\delta(\alpha^*)}(\phi) \tag{3}$$

This can be concisely summarized in Algorithm 1, although we remind the reader that $\mathcal{L}(\cdot)$ is defined with respect to replay buffer $\mathcal{D}$ and is not exactly equivalent to optimizing $J(\cdot)$. Unless specified, we by default use consistent hyperparameters (found in Appendix F) for all comparisons found inside a figure, although learning rate and minibatch size may be altered when training models of different sizes due to GPU memory limits. Thus, even though RL is commonly sensitive to hyperparameters (Zhang et al., 2021), we find that **once a pre-existing RL baseline has already been setup, incorporating DARTS requires no extra cost in tuning, as evidence of its ease-of-use.**

## 4 EXPERIMENTS

### 4.1 EXPERIMENT SETUP

We perform key experiments to understand the behavior of RL-DARTS, and answer the following main questions in our analysis:

1. Does the supernet train end-to-end at all or will it fail to even initially collect good online data? Does $\alpha$ converge towards a sparse solution over time, and what is the computational cost?
2. Even if the supernet trains, do the corresponding discrete cells also improve in evaluation performance throughout $\alpha$'s training, despite common integrality gap issues? What kinds of failure modes occur?
3. How do the final discrete cells perform at evaluation, and how does RL-DARTS compare against random search? Can a large-scale joint optimization over multiple diverse environments lead to a single transferrable cell?

In order to study the behavior of RL-DARTS, we use the Procgen benchmark (Cobbe et al., 2019; 2020), which readily provides pre-existing baselines for both PPO (Schulman et al., 2017) and Rainbow DQN (Hessel et al., 2018) algorithms, over a diverse selection of 16 games, each with infinite levels to benchmark RL-DARTS in large data regimes where episodes may drastically change, relevant for generalization. We use the standard "easy" difficulty commonly benchmarked by a variety of other works (Raileanu et al., 2020; Raileanu & Fergus, 2021; Parker-Holder et al., 2021b).

We also use the default IMPALA-CNN architecture (Espeholt et al., 2018) as a strong hand-designed baseline when comparing both supernet and discrete cell performances. IMPALA-CNN can be seen as specific instance of the stacked cell design in Subsection 3.1 and Figure 2, where its "Reduction Cell" consists of a Conv3x3 and MaxPool3x3 (Stride 2) with $R = 1$ and its "Normal Cell" consists of a residual layer with Conv3x3's and ReLU's, with $N = 2$. More details can be found in the original paper (Espeholt et al., 2018).

As for RL-DARTS's search space, we consider the following base ops $\mathcal{O}_{base}$, corresponding algorithms, and $(N, R, I)$:

- **Classic + PPO:** $\mathcal{O}_{base,N}$ = {Conv3x3+ReLU, Conv5x5+ReLU, Dilated3x3+ReLU, Dilated5x5+ReLU} for normal ops and $\mathcal{O}_{base,R}$ = {Conv3x3, MaxPool3x3, AveragePool3x3} for reduction ops, which is standard in supervised learning (Liu et al., 2019; Zoph et al., 2018; Pham et al., 2018). We use $(N, R, I) = (1, 1, 4)$ to be comparable to the baseline.
- **Micro + Rainbow:** $\mathcal{O}_{base,N}$ = {Conv3x3, ReLU, Tanh}, a more fine-grained and novel search space which has not been used previously in SL. The inclusion of Tanh is motivated by its use previously for continuous control architectures (Salimans et al., 2017; Song et al., 2020). In order to avoid hidden confounding effects from the reduction cell when displaying the normal cell in figures, we use $(N, R, I) = (2, 0, 4)$, where reduction ops default to IMPALA-CNN's.

### 4.2 TRAINING THE SUPERNET

In SL, it is expected to see the supernet's asymptotic validation accuracy eventually reach a sufficient threshold, comparable to (although slightly worse than) a discrete cell's validation accuracy (Liu et al., 2019; Li et al., 2020). The question is, can an RL supernet achieve the same result for training reward? As mentioned earlier in the introduction, this is unclear as a negative feedback loop can

occur where initially, a suboptimal supernet collects poor data for training, which leads to an even worse supernet, as $\alpha$ may potentially upweight suboptimal ops. However, we find that **training the supernet end-to-end works effectively, even with minimal hyperparameter tuning** for both PPO (Figure 3) and Rainbow (Figure 4), although PPO may take longer (50M steps) to reach comparable scores to IMPALA-CNN while Rainbow only needs 25M steps. However, this is a non-issue during discretization, as the $\alpha$ has already converged to a sparse solution (via softmax weight magnitudes) by 25M steps on PPO.

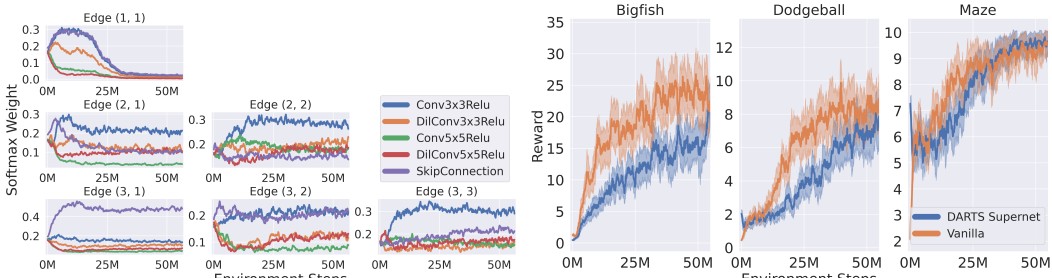

**Figure 3: Left:** Softmax op weights over all edges in the cell when training the supernet with PPO + "Classic" search space on Dodgeball. Zero op weight is not shown to improve clarity. Note that by 25M steps, the op choices have already converged towards a sparser solution. **Right:** Sanity check to verify that the supernet achieves a regular training curve, using vanilla IMPALA-CNN as a rough gauge. Both use depths $16 \times 3$.

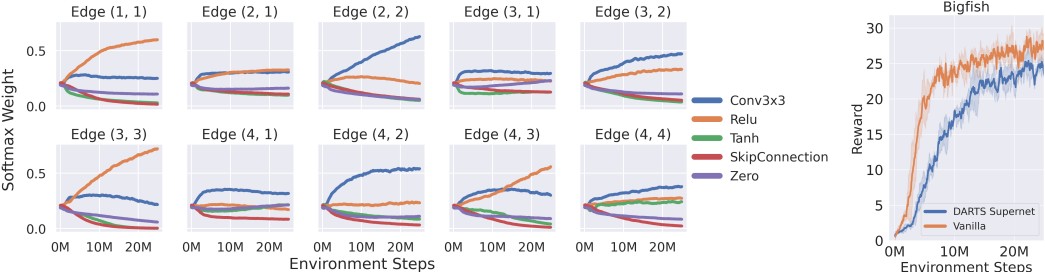

**Figure 4:** Analogous settings with Figure 3 using Rainbow + "Micro" search space. **Left:** Softmax weights when training Rainbow with infinite levels on Bigfish, also converging towards a sparser solution. **Right:** Sanity check for supernet when using Rainbow.

Interestingly, on Dodgeball with PPO, we find that the process strongly downweights all base ops except for the standard Conv3x3+ReLU, with 5x5 ops possessing the lowest softmax weights. Conveniently, this provides an opportunity to understand whether $\alpha$ **downweights suboptimal ops throughout training.** We confirm this result in Table 1 by performing evaluations

| Scenario | Conv 3x3 | Conv 5x5 |
|---|---|---|
| Train (Inf. levels) | $\mathbf{15.1 \pm 2.5}$ | $13.2 \pm 2.3$ |
| Train (200 levels) | $\mathbf{12.1 \pm 1.7}$ | $9.8 \pm 2.1$ |
| Test (from Train) | $\mathbf{10.2 \pm 2.3}$ | $5.9 \pm 1.7$ |

**Table 1:** PPO IMPALA-CNN evaluations (mean return at 50M steps) on Dodgeball. Learning curves can be found in Appendix C.1, Figure 13.

over standard IMPALA-CNN cells using either purely 3x3 or 5x5 convolutions for the whole network, and demonstrating that the 3x3 setting outperforms the 5x5 setting (especially in limited data, e.g. 200-level training/test regime), suggesting the signaling ability of $\alpha$ on op choice. Although the

| | Rainbow (25M Steps) | | PPO (50M Steps) | |
|---|---|---|---|---|
| Scenario | Trainable $\alpha$ | Uniform $\alpha$ | With ReLU | No ReLU |
| Training (Inf. levels) Reward | $\mathbf{3.1 \pm 0.5}$ | $0.9 \pm 0.2$ | $\mathbf{7 \pm 0.9}$ | $1.9 \pm 0.3$ |

**Table 2:** Supernet training rewards on Dodgeball. Learning curves can be found in Appendix B.

training abilities of supernets in RL have been established, an important contrasting question now surfaces: *Can any supernet train?* The supernet possesses an incredibly dense set of weights, and thus one might wonder whether trainability occurs with any search space or settings. We answer in the negative, where a **poorly designed supernet can fail.** To show this clearly, we remove all ReLU nonlinearities from the "Classic" search space used for PPO (although ReLUs still exist in final MLP projections), as well as simply freeze $\alpha$ to be uniform for Rainbow, and find both cases produce poor training in Table 2. **Thus, the supernet in RL provides important search signals in terms of reward and $\alpha$ during training, especially on the quality of a search space.**

We further provide computational efficiency metrics in Table 3, where we find that the practical wall-clock time required for training the supernet (i.e. the search cost) is very comparable with DARTS in SL (Liu et al., 2019), requiring only a few GPU days. We do note that unlike SL where the vast majority of the cost is due to the network, RL time cost is partially based on non-network factors such as environment simulation, and thus wall-clock times may change depending on specific implementation.

| Network | Training Cost in GPU Days (w/ specific algorithm) |
|---|---|
| IMPALA-CNN | 1 (PPO), 0.5 (Rainbow) |
| "Classic" Supernet | 2.5 (PPO) |
| "Micro" Supernet | 1.5 (Rainbow) |
| CIFAR-10 Supernet (Liu et al., 2019) | 4 (SL/Original DARTS) |

**Table 3:** Computational efficiency in terms of wall-clock time, achieved on a V100 GPU. For the RL cases (PPO + Rainbow), all networks use depths of $16 \times 3$. Training cost in RL is defined as the wallclock time taken to reach 25M steps, rounded to the nearest 0.5 GPU day. We have also included reported time for DARTS in SL (Liu et al., 2019) as comparison.

### 4.3 DISCRETE CELL IMPROVEMENT

Due to the existence of *integrality gaps* in SL between a continuously relaxed supernet and its discrete cell (Wang et al., 2021b;a), this also raises the same question in RL. We demonstrate that even using the default $\delta$ discretization from (Liu et al., 2019) leads to both qualitative and quantitative improvements, although we leave potential improvements to future works.

In terms of **qualitative improvement**, we display discretized cells when training a supernet with Rainbow on the Starpilot environment in Figure 5. As we can see, the earlier cell consists of only linear operators is clearly a poor design in comparison to the later cell. We provide more extensive cell evolution ablations for PPO in Figures 14, 15 in Appendix C.2, displaying more sophisticated yet still interpretable changes in cell topology.

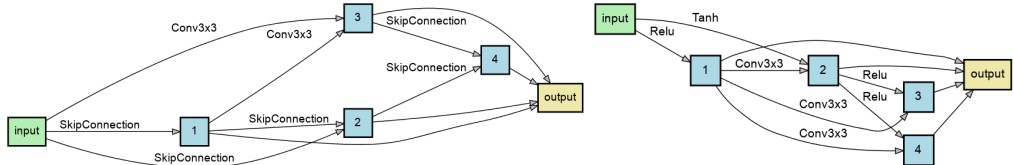

**Figure 5:** Evolution of discovered cells over a DARTS optimization process. **Left**: A cell discovered in the early stage which is dominated by skip connections and only linear ops. **Right**: A cell discovered in the end which possesses several reasonable local structures similar to Conv + ReLU residual connections.

For **quantitative improvement**, as $\alpha$ changes during supernet training, so do the outputs of the discretization procedure on $\alpha$. We collect all distinct discrete cells $\{\delta(\alpha_1), \delta(\alpha_2), \ldots\}$ into a sequence, and evaluate each cell's performance $\max_\phi J(\pi_{\phi,\delta(a_i)}) \; \forall i$ via training from scratch for 25M steps, displayed in Figure 6. As we can see, the performance generally improves over time, indicating that supernet optimization selects better cells. However, we find that such behavior can be environment-dependent, as some environments possess less monotonic evaluation curves (see Figure 16 in Appendix C).

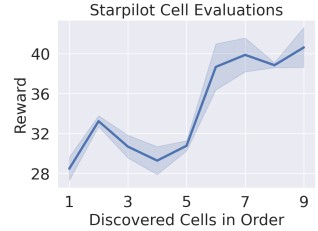

**Figure 6:** Evaluation of 9 distinct discrete cells in order from the trajectory of $\alpha$ on the Starpilot environment when using Rainbow.

One reason why a **discretized cell may underperform is if its corresponding supernet fails to learn**. One simple reason is due to the inherent sensitivity and variance in RL training. For example, in Figure 7, we see that when using Rainbow, bad training setups (ex: setting $D$ too high) can create an unlearnable "Supernet 2", subsequently leading to an unlearnable corresponding discrete cell, while the inverse occurs with "Supernet 1" and its own discrete cell. The discrete cells' rewards can also be qualitatively explained (ex: see Figure 7's caption). We perform further studies in Appendix C.3, Figure 17, showing that supernet and discrete cell rewards are indeed correlated, but also have environment-dependent integrality gaps, suggesting that search quality can be improved via both

better supernet training (Raileanu et al., 2020; Zhang et al., 2021), as well as better discretization procedures (Liang et al., 2019; Wang et al., 2021b).

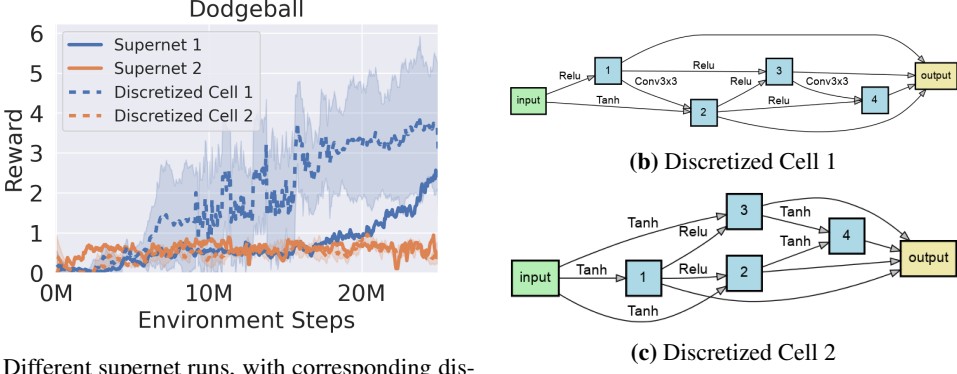

**(a)** Different supernet runs, with corresponding discretized cell (depths $64 \times 5$) training curves.

**(b)** Discretized Cell 1

**(c)** Discretized Cell 2

**Figure 7: (a)** Two different supernets trained on the Dodgeball environment using Rainbow, with corresponding discretized cells evaluated using 3 random seeds. **(b)** Discretized cell from Supernet 1. Note the similarity to regular Conv3x3 + ReLU designs. **(c)** Discretized cell from Supernet 2, which uses too many Tanh nonlinearities, known to cause vanishing gradient effects.

## 4.4 SINGLE GAME EVALUATIONS

Since the results above establish RL-DARTS's search capability, we now focus on comparisons between discrete architectures and baselines. We first present results in the standard scenario, where RL-DARTS is applied to each game independently, discretized at 25M steps.

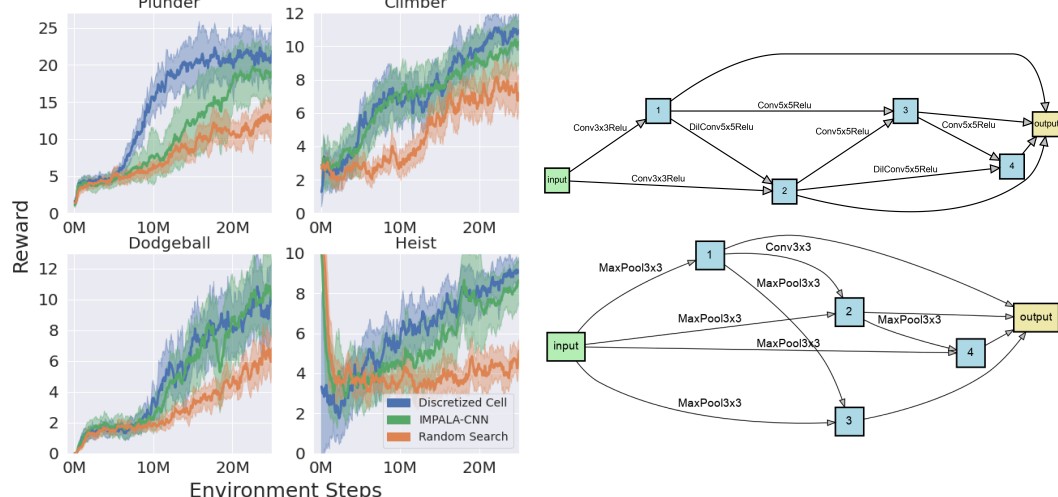

**Figure 8: Left 2x2 Plot:** Examples of discrete cell evaluations using the "Classic" search space with PPO, with depths $64 \times 3$. **Right:** Normal (Top) and Reduction (Bottom) cells found for "Plunder" which achieves faster training than IMPALA-CNN. Note the interesting use of 5x5 convolutional kernel sizes later in the cell.

| | IMPALA-CNN | RL-DARTS (Discrete Cell) | Random Search |
|---|---|---|---|
| Avg. Normalized Reward | 0.708 | 0.709 | 0.489 |

**Table 4:** Average normalized rewards across all 16 environments w/ PPO, using the normalization method from (Cobbe et al., 2020). Full details and results (including Rainbow) are presented in Appendix D.

We use random search as an important baseline to gauge the search space's difficulty (ex: "Classic" total size is $4 \times 10^{11}$, see Appendix G.1). For fair comparison, we ensure total wall-clock time (with same hardware) stays equal, as common in (Liu et al., 2019). Since in Table 3, a PPO supernet takes 2.5x longer to reach 25M steps, this is rounded to a random search budget of 3 cells to be trained with depths $16 \times 3$ for 25M steps. The best

of the 3 cells is used for full evaluation. In Table 4, on average, random search under-performs significantly, which also shows IMPALA-CNN is a strong hand-designed baseline.

However, in Figure 8, **RL-DARTS is capable of finding architectures from scratch which outperform IMPALA-CNN on few environments such as Plunder and Heist, while maintaining competitive performance on others.** On the Rainbow side, in Figure 9, we found that in certain cases, **even 100 unique random cells underperform against RL-DARTS!**

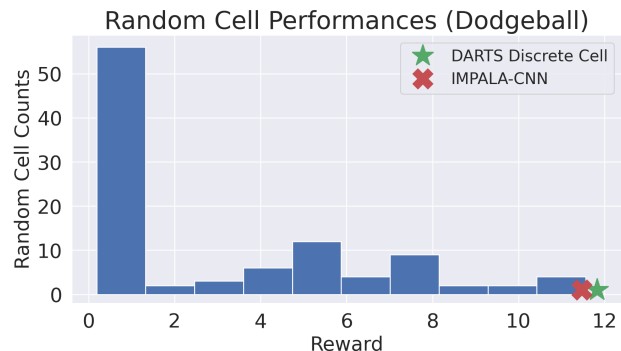

### 4.5 TRANSFER EVALUATION

Instead of performing the RL-DARTS by joint training a supernet across infinite levels within only *one game*, one

**Figure 9:** Histogram of 100 random cells' rewards on Dodgeball with Rainbow + "Micro" search space (depths $64 \times 3$), with 95% substantially worse and 55% unable to even train. More results in Appendix E.

may ask if it is possible to **joint train a supernet across *all games* over all infinite levels to find a single transferrable cell**, which may be of interest to practitioners seeking to find a common architecture general to a broad class of RL problems. We confirm this approach using Rainbow, where a learner performs gradient updates over actor replay data (with normalized rewards) from all 16 games.

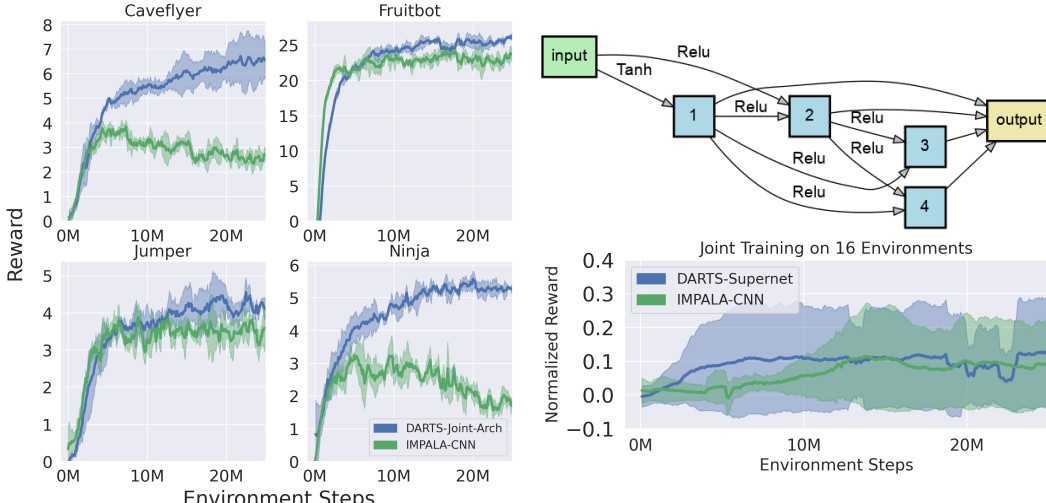

**Figure 10: Left 2x2 Plot:** Evaluation of the discrete cell joint-trained over all environments using depths $64 \times 5$ to emphasize comparison differences. **Right (Top):** Discrete cell found. **Right (Bottom):** Average Normalized rewards over all 16 games during supernet + baseline training.

## 5 CONCLUSION

Our research has demonstrated that in a minimally invasive way, it is entirely possible to apply DARTS to RL, a completely different optimization paradigm than SL, which DARTS was originally proposed for. We believe this work can open new doors to both NAS and RL communities alike. Especially for theorists and analysts of DARTS, RL is a completely new frontier for which to understand softmax routing and continuous relaxation techniques. Previous DARTS techniques (Wang et al., 2021b; Chu et al., 2020a; Liang et al., 2019) may be tested in the RL setting, and new RL-specific ways of discretization and training may also be proposed, increasing RL-DARTS's utility even more. For RL practitioners, RL-DARTS could be important to finding better and more efficient (Cai et al., 2019) architectures for large-scale robotics (James et al., 2019; Akinola et al., 2021), transferable architectures in offline RL (Levine et al., 2020), as well as RNNs for memory (Pritzel et al., 2017; Fortunato et al., 2019; Kapturowski et al., 2019) and adaptation (Duan et al., 2016; Wang et al., 2017).

## 6 STATEMENTS

**Ethics Statement:** The only relevant potential concerns in our work occur normally with general NAS methods, which can sacrifice model interpretability in order to achieve higher objectives. For the field of RL specifically, this may warrant more attention in AI safety when used for real world robotic pipelines. Furthermore, as with any NAS research, the initial phase of discovery and experimentation may contribute to carbon emissions due to the computational costs of extensive tuning. However, this is usually a means to an end, such as an efficient search algorithm, which this paper proposes with no extra hardware costs.

**Reproducibility Statement:** We base our statement on the "NAS Best Practices Checklist" (Lindauer & Hutter, 2019), recently released to improve reproducibility in neural architecture search.

In terms of raw reproducibility, we discuss the explicit hyperparameters in Appendix F, as well as search space details in Subsection 4.1. The original DARTS implementation is readily available across Github (e.g. `https://github.com/quark0/darts`) as it is a highly popular algorithm. Similarly, Procgen is publicly available (`https://github.com/openai/procgen`) along with its training code and details (Cobbe et al., 2020). For our case, we used TF-Agents for PPO, available at `https://www.tensorflow.org/agents` and Acme for Rainbow, available at `https://github.com/deepmind/acme`. Since our algorithm is relatively simple to assemble given preexisting pipelines, we believe that our approach should require minimal effort to reproduce.

In terms of comparing NAS methods, we indeed controlled for confounding factors by using the same hardware, as shown in Table 3. For ablation studies, we provided a large set of studies as seen throughout Subsections 4.2 and 4.3 as well as in the Appendix. We also ensured the same evaluation protocol via making sure all depths remain the same (16 for supernets, 64 for discrete cells, and $D = 3$ layers for cheaper large-scale runs or $D = 5$ for fine-grained A/B testing). In terms of performance comparisons over time, we showed the performance of RL-DARTS's discrete cells over the search procedure / supernet training in Figure 6. We extensively covered the comparisons against random search in Subsection 4.4, over all 16 games as well as a large scale 100 random cell comparison in Figure 9. For seeded runs, as standard in RL, we performed 3-seeded runs for all experiments.

In terms of hyperparameter tuning, we discuss this in Appendix F and also in Subsection 3.2. The time reported for all components of the NAS method can be found in Table 3, and we explicitly reported all of the details of our experiments throughout the paper.

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

# APPENDIX

## A  REBUTTAL

### A.1  TRAINING PROCEDURE (EXTENDED)

Below are PPO (Schulman et al., 2017) and Rainbow-DQN (Hessel et al., 2018) RL-DARTS variants. Exact loss definitions and data collection procedures can be found in their respective papers.

---

**Algorithm 2:** RL-DARTS with PPO

---

Supernet training:

  Setup supernet encoder $f_{\theta_e,\alpha}$ with weights $\theta_e$.
  Initialize policy and value head projection weights $W_\pi \in \mathbb{R}^{d,|\mathcal{A}|}, W_v \in \mathbb{R}^{d,1}$.
  Collect all trainable weights $\theta = \{\theta_e, W_\pi, W_v\}$.
  Setup policy $\pi_{\theta,\alpha}(s) \sim \text{softmax}(W_\pi \cdot f_{\theta_e,\alpha}(s))$.
  Setup value function $V_{\theta,\alpha}(s) = W_v \cdot f_{\theta_e,\alpha}(s)$.
  Define standard PPO loss $\mathcal{L}(\theta, \alpha)$ using $\pi_{\theta,\alpha}$ and $V_{\theta,\alpha}$.
  Perform PPO training via collecting data from $\pi_{\theta,\alpha}$ and SGD with $\nabla_{\theta,\alpha}\mathcal{L}(\theta, \alpha)$.
  Collect $\alpha^*$ from previous training procedure.

Discretization:

  Let $\delta(\alpha^*)$ be the discrete cell constructed via Algorithm 4.

Evaluation:

  Setup discretized cell encoder $f_{\phi_e,\delta(\alpha^*)}$.
  Initialize policy and value head projection weights $W'_\pi \in \mathbb{R}^{d,|\mathcal{A}|}, W'_v \in \mathbb{R}^{d,1}$.
  Collect all trainable weights $\phi = \{\phi_e, W'_\pi, W'_v\}$.
  Setup policy $\pi_{\phi,\delta(\alpha^*)}(s) \sim \text{softmax}(W'_\pi \cdot f_{\phi_e,\delta(\alpha^*)}(s))$.
  Setup value function $V_{\phi,\delta(\alpha^*)}(s) = W'_v \cdot f_{\phi_e,\delta(\alpha^*)}(s)$.
  Define standard PPO loss $\mathcal{L}_{\delta(\alpha^*)}(\phi)$ using $\pi_{\phi,\delta(\alpha^*)}$ and $V_{\phi,\delta(\alpha^*)}$.
  Perform PPO training via collecting data from $\pi_{\phi,\delta(\alpha^*)}$ and SGD with $\nabla_\phi \mathcal{L}_{\delta(\alpha^*)}(\phi)$.
  Report final policy reward.

---

**Algorithm 3:** RL-DARTS with Rainbow. Note that we do not use noisy nets in this implementation.

---

Supernet training:

  Setup supernet encoder $f_{\theta_e,\alpha}$ with weights $\theta_e$.
  Initialize dueling network projections $W_v \in \mathbb{R}^{d,1}, W_a \in \mathbb{R}^{d,|\mathcal{A}|}$.
  Collect all trainable weights $\theta = \{\theta_e, W_v, W_a\}$.
  Setup value network $V_{\theta,\alpha}(s) = W_v \cdot f_{\theta_e,\alpha}(s)$.
  Setup advantage network $A_{\theta,\alpha}(s,a) = W_a \cdot f_{\theta_e,\alpha}(s)$.
  Setup Q-network $Q_{\theta,\alpha}(s,a) = V_{\theta,\alpha}(s) + A_{\theta,\alpha}(s,a) - \frac{1}{|\mathcal{A}|}\sum_{a'\in\mathcal{A}} A_{\theta,\alpha}(s,a')$.
  Define standard Rainbow loss $\mathcal{L}(\theta, \alpha)$ using $Q_{\theta,\alpha}$.
  Perform Rainbow training via collecting data from $Q_{\theta,\alpha}$ and SGD with $\nabla_{\theta,\alpha}\mathcal{L}(\theta, \alpha)$.
  Collect $\alpha^*$ from previous training procedure.

Discretization

  Let $\delta(\alpha^*)$ be the discrete cell constructed via Algorithm 4.

Evaluation

  Setup discretized cell encoder $f_{\phi_e,\delta(\alpha^*)}$ with weights $\phi_e$.
  Initialize dueling network projections $W'_v \in \mathbb{R}^{d,1}, W'_a \in \mathbb{R}^{d,|\mathcal{A}|}$
  Collect all trainable weights $\phi = \{\phi_e, W'_v, W'_a\}$
  Setup value network $V_{\phi,\delta(\alpha^*)}(s) = W'_v \cdot f_{\phi_e,\delta(\alpha^*)}(s)$
  Setup advantage network $A_{\phi,\delta(\alpha^*)}(s,a) = W'_a \cdot f_{\phi_e,\delta(\alpha^*)}(s)$
  Setup Q-network
  $Q_{\phi,\delta(\alpha^*)}(s,a) = V_{\phi,\delta(\alpha^*)}(s) + A_{\phi,\delta(\alpha^*)}(s,a) - \frac{1}{|\mathcal{A}|}\sum_{a'\in\mathcal{A}} A_{\phi,\delta(\alpha^*)}(s,a')$
  Define standard Rainbow loss $\mathcal{L}_{\delta(\alpha^*)}(\phi)$ using $Q_{\phi,\delta(\alpha^*)}$.
  Perform Rainbow training via collecting data from $Q_{\phi,\delta(\alpha^*)}$ and SGD with $\nabla_\phi \mathcal{L}_{\delta(\alpha^*)}(\phi)$.
  Report final policy reward.

---

**Algorithm 4:** Discretization Procedure.

---

Argmax:

    For $(i, j)$ across all edges:

        Define edge strength $w_{i,j} = \max_{o \in \mathcal{O}, o \neq zero} p_o^{(i,j)}$.

        Define edge op $o_{(i,j)} = \arg\max_{o \in \mathcal{O}, o \neq zero} p_o^{(i,j)}$.

Prune:

    For node $j$ in all intermediate nodes:

        Sort input edge weights $w_{i_1,j} \geq w_{i_2,j} \geq \ldots$

        Retain only top $K$ edges $(i_1, j), \ldots, (i_K, j)$ and corresponding ops $o_{(i_1,j)}, \ldots, o_{(i_K,j)}$ in
           final cell.

---

Note that both RL-DARTS procedures can also be summarized in terms of raw code as simple one-line edits to the image encoder used (compressing the rest of the regular RL training pipeline code):

```
def train(feature_encoder):
    """Initial RL algorithm setup"""
    ...
    extra_variables = Wrap(feature_encoder)
    all_trainable_variables =
        [feature_encoder.trainable_variables(), extra_variables]
    """Rest of RL algorithm setup"""
    ...
    apply_gradients(loss, all_trainable_variables)
```

Thus, the 3-step RL-DARTS procedure from Section 3.2 can be seen as:

```
DARTSSuperNet = MakeSuperNet(ops, num_nodes) # Setup
train(DARTSSuperNet) # Supernet training
DiscretizedNet = DARTSSuperNet.discretize() # Discretization
train(DiscretizedNet) # Evaluation
```

## A.2 POSSIBLE IMPROVEMENTS TO RL-DARTS

These include:

1. Discretization Changes: One may consider discretization based on the total reward $J(\pi_{\theta,\alpha})$, which may provide a better signal for the correct discrete architecture. This is due to the fact that the relative strengths of operation weights from $\alpha$ may not correspond to the best choices during discretization. (Wang et al., 2021b) considers iteratively pruning edges from the supernet based on maximizing validation accuracy changes. For RL, this would imply a variant of discretization dependent on multiple calculations of $J(\pi_{\theta^*, \delta_1(\alpha^*)}) - J(\pi_{\theta^*, \delta_2(\alpha^*)})$ where $\theta^*$ consists the weights obtained during supernet training, as well as fine-tuning $J(\pi_{\theta^*, \delta_1(\alpha^*)})$ at every pruning step. These changes, in addition to the inherently noisy evaluations of $J(\cdot)$, greatly increase the complexity of the discretization procedure, but are worth exploring in future work.

2. Changing the Loss / Regularization: Throughout this paper, we have found that vanilla DARTS is able to train by simply optimizing $\alpha$ with respect to the loss, even though in principle, the loss is not strongly correlated to the actual reward in RL. Thus, it is curious to understand whether loss-based metrics or modifications may help improve RL-DARTS. One such modification is based on the observation that certain RL losses may not be required for training $\alpha$. In PPO, the entropy loss of $\pi_\theta$ may not be necessary or useful for improving the search quality of $\alpha$, and thus it may be better to perform a two step update by providing a different loss for $\alpha$. One may also consider searching for two separate encoders via two supernets, since both PPO and Rainbow feature separate networks, e.g. the policy $\pi_{\theta_1, \alpha_1}$ and value function $V_{\theta_2, \alpha_2}$ for PPO and advantage function $A_{\theta_1, \alpha_1}$ and value function $V_{\theta_2, \alpha_2}$ for Rainbow.

3. Signaling Metrics and Early Stopping: Observing metrics throughout training allows for early stopping, which can reduce search cost and provide better discrete cells. This includes metrics such as the strength of certain operation weights (Liang et al., 2019) as well as the Hessian with respect to $\alpha$ throughout training, i.e. $\nabla_\alpha^2 \mathcal{L}(\theta, \alpha)$ as found in (Zela et al., 2020). Furthermore, inspired by performance prediction methods (Mellor et al., 2020; Luo et al., 2018), one may analyze metrics such as the Jacobian Covariance, via the score defined to be the $-\sum_{i=1}^{B} \left[ \log(\sigma_i + \varepsilon) + (\sigma_i + \varepsilon)^{-1} \right]$ where $\varepsilon = 10^{-5}$ is a stability constant and $\sigma_1 \leq \ldots \leq \sigma_B$ are the eigenvalues of the correlation matrix corresponding to the Jacobian $J = \left[ \frac{\partial f}{\partial s_1}, \ldots, \frac{\partial f}{\partial s_B} \right]^T$ with $B$ input images $\{s_1, \ldots, s_B\}$.

This metric/predictor has been found to be a strong signal for accuracy in SL NAS among many previous predictors (White et al., 2021). However, for the RL case, just like the loss, the mentioned metrics must be defined with respect to the current replay buffer $\mathcal{D}$, and thus raises the question of what type of data is to be used for calculating these metrics. When using a reasonable variant where the data is collected from a pretrained policy, we find that methods such as Jacobian Covariance do not provide meaningful feedback as shown in Figure R1.

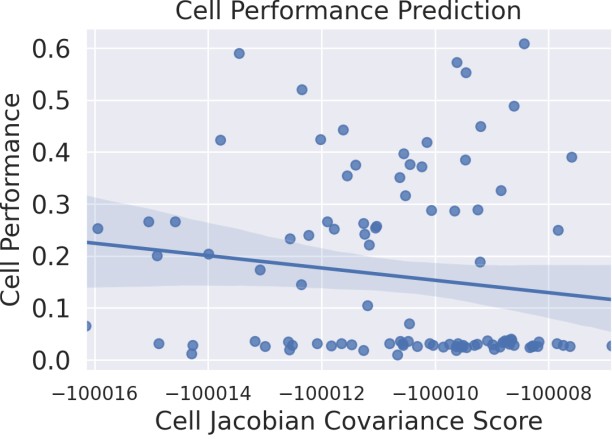

**Figure R1:** Attempts to predict random cell performances on the Dodgeball environment by using the Jacobian covariance predictor. Jacobian covariance scores are computed as in (Mellor et al., 2020) by evaluating the networks (using the cells with depths $64 \times 3$) on mini-batches of $B = 16$ random images sampled from the Dodgeball environment using a pretrained policy's trajectories. For each cell, Jacobian Covariance scores across mini-batches are averaged, and the performance on the Dodgeball environment is normalized by the maximum possible return.

4. Supernet Training: As seen from the results in Figure 7 (main body) and Appendix C.3, there is a correlation between supernet and discrete cell performances. However, this is affected by the environment used as well as integrality gaps between the continuous relaxation and discrete counterparts, and thus further exploration is needed before concluding that improving the supernet training leads to better discrete cell performances. In any case, reasonable methods of improving the supernet can involve DARTS-agnostic modifications to the RL pipeline, including data augmentation (Kostrikov et al., 2020; Raileanu et al., 2020) as well as online hyperparameter tuning (Parker-Holder et al., 2020; 2021a). Simple hyperparameter tuning (e.g. on the softmax temperature for calculating $p_o^{(i,j)}$'s) also can be effective.

## A.3 DM-Control with SAC (Extra Benchmark) Results

We also include results via using a Soft Actor Critic (SAC) (Haarnoja et al., 2018) variant of RL-DARTS on DM-Control (Tassa et al., 2018) with image-based observations resized to $64 \times 64$ with a frame-stacking of 3. The code can be found in `https://github.com/google-research/pisac`, where we disabled the predictive information loss to use only regular SAC. The baseline architecture is a 4-layer convolutional architecture found in `https://github.com/google-research/pisac/blob/master/pisac/encoders.py`. Both

our DARTS supernet and discrete cells use $N = 3, I = 4, K = 1$ using the "Micro" search space, with convolutional depths of 32 to remain fair to the baseline.

Our results in Figure R2 thus also provide evidence of RL-DARTS's flexibility with yet a 3rd algorithm (SAC), as well as on continuous control, where discrete cells outperform the baseline.

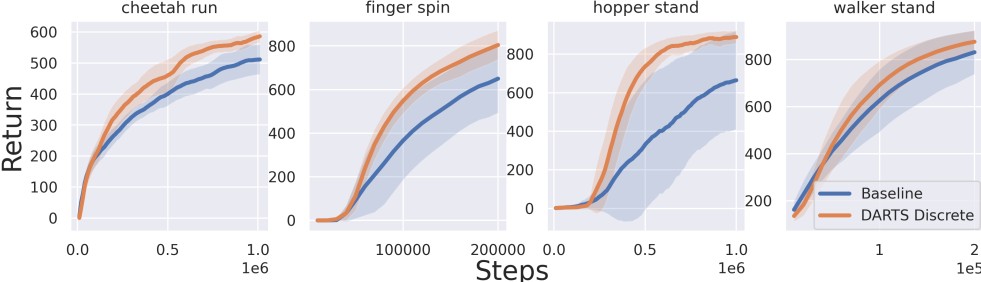

**Figure R2:** Results with RL-DARTS discrete cells on DM-Control when using the SAC algorithm.

# B    WHAT AFFECTS SUPERNET TRAINING?

Given the positive training results we demonstrated in the main body of the paper, one may wonder, *can any supernet, no matter how poorly designed or setup, still train well in the RL setting?* If so, this would imply that the search method would not be producing meaningful, but instead, random signals.

We refute this hypothesis by performing ablations over our supernet training in order to have a better understanding of what components affect its performance. We ultimately show that the search space and architecture variables play a very significant role in its optimization, thus validating our method.

## B.1    ROLE OF SEARCH SPACE

We remove the ReLU nonlinearities from the "Classic" search space, so that $\mathcal{O}_{base}$ = {Conv3x3, Conv5x5, Dilated3x3, Dilated5x5} and thus the DARTS cell consists of only linear operations. As shown in Fig. 11, this leads to a dramatic decrease in supernet performance, providing evidence that the search space matters greatly.

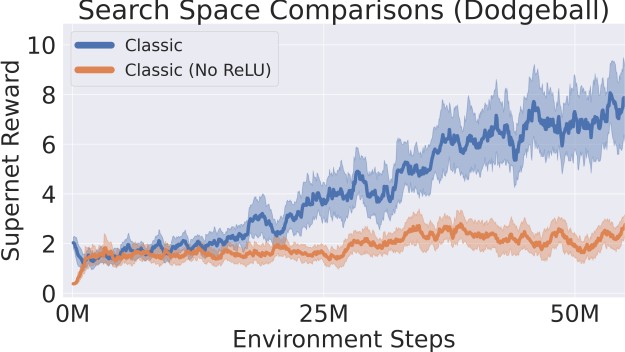

**Figure 11:** Supernet training using PPO on Dodgeball with infinite levels, when using the "Classic" search space with/without ReLU nonlinearities, under the same hyperparameters.

## B.2    UNIFORM ARCHITECTURE VARIABLES

We further demonstrate the importance of the architecture variables $\alpha$ on training. We see that in Fig. 12, freezing $\alpha$ to be uniform throughout training makes the Rainbow agent unable to train at all. This suggests that it is crucial for $\alpha$ to properly route useful operations throughout the network.

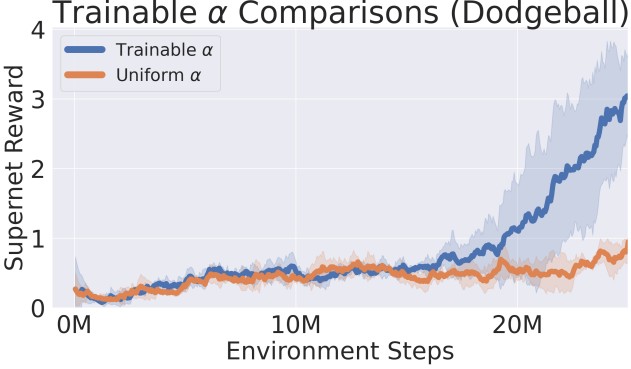

**Figure 12:** Supernet training using Rainbow on Dodgeball with infinite levels, when using the "Micro" search space with/without trainable architecture variables $\alpha$, under the same hyperparameters.

# C    WHAT AFFECTS DISCRETE CELL PERFORMANCE?

## C.1    SOFTMAX WEIGHTS VS DISCRETIZATION

As seen from Figure 3 in the main body of the paper, the DARTS supernet strongly downweights Conv5x5+ReLU operations when using the "Classic" search space with PPO. In order to verify

the predictive power of the softmax weights, as a proxy, we thus also performed evaluations when using purely 3x3 or 5x5 convolutions on a large IMPALA-CNN with $64 \times 5$ depths. We see that the Conv3x3 setting indeed outperforms Conv5x5, corroborating the results in which during training, $\alpha$ strongly upweights the Conv3x3+ReLU operation and downweights Conv5x5+ReLU.

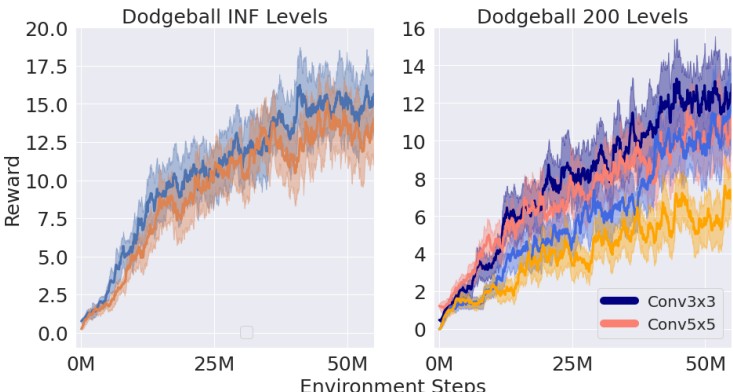

**Figure 13:** Large IMPALA-CNNs evaluated on Dodgeball using either Infinite or 200 levels with PPO. For the 200 level setting, lighter colors correspond to test performance.

## C.2 DISCRETE CELL EVOLUTIONS

Along with Figure 5 in the main body of the paper, we also compare extra examples of discretizations before and after supernet training, to display reasonable behaviors induced by the trajectory of $\alpha$.

In Figure 14, we use PPO with the "Classic" search space, but instead use $(N, R, I) = (1, 0, 6)$ along with outputting the last node (instead of concatenation with a Conv1x1 for the output) to allow a larger normal cell search space and graph topology. In Figure 15, the discretized cell initially uses a large number of skip connections as well as dead-end nodes. However, at convergence, it eventually utilizes all nodes to compute the final output. Curiously, we find that the skip connection between the input and output appears commonly throughout many searches.

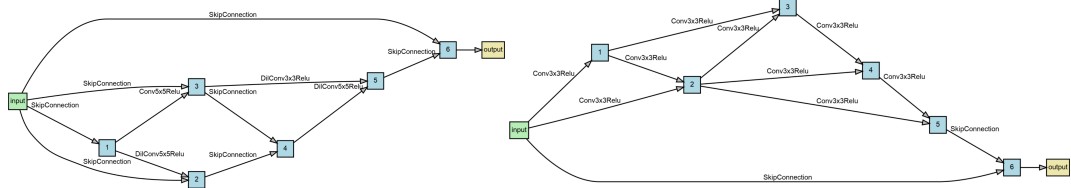

**Figure 14:** Comparison of discretized cells before and after supernet training, on Starpilot using PPO with $I = 6$ nodes.

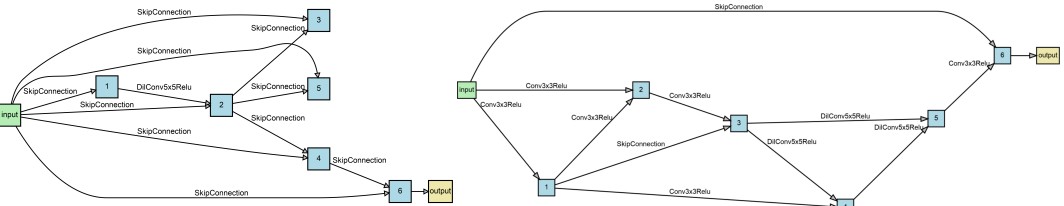

**Figure 15:** Comparison of discretized cells before and after supernet training, on Plunder using PPO with $I = 6$ nodes.

For the Rainbow setting, in Figure 6 in the main body of the paper, we saw that when the search process is successful, the supernet's training trajectory induces discretized cells which improve evaluation performance as well. The cells discovered later generally perform better than cells discovered earlier in the supernet training process. In Figure 16, we show more examples of such evaluation curves.

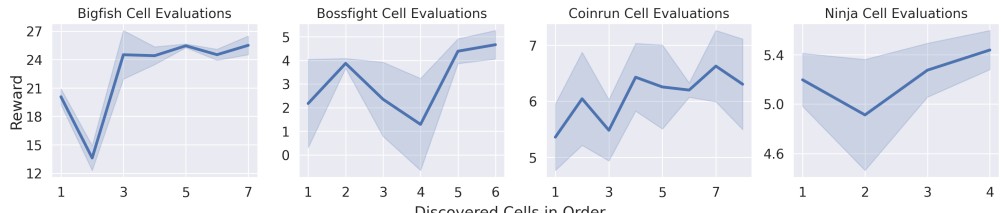

**Figure 16:** Evaluated discretized cells discovered throughout training the supernet with Rainbow. To save computation, we evaluate every 2nd cell that was discovered.

### C.3 CORRELATION BETWEEN SUPERNET AND DISCRETIZED CELLS

Given that the discretized cell only explicitly depends on architecture variables $\alpha$ and not necessarily model weights $\theta$, one may wonder: *Is there a relationship between the rewards of the supernet and of its corresponding discretized cell?* For instance, the degenerate/underperformance setting mentioned in Section 4.4 and Appendix C can be thought of as an extreme scenario. At the same time, there could be an integrality gap, where there the discretization process $\delta(\alpha)$ produces cells which give different rewards than the supernet.

In order to make such an analysis comparing rewards, we first must prevent confounding factors arising from Rainbow's natural performance on an environment regardless of architecture. We thus first divide the supernet and discrete cell scores by the score obtained by the IMPALA-CNN baseline, where the baseline and discrete cells all used depths of $64 \times 3$.

In Figure 17, when using Rainbow and observing across environments, we find both high correlation and also integrality gaps: for some environments such as Ninja and Coinrun, there is a significant correlation between supernet and discrete cell rewards, while for other environments such as Dodgeball, there is a significant gap. This suggests that search quality can be improved via both better supernet training such as using hyperparameter tuning or data augmentation (Raileanu et al., 2020), as well as better discretization procedures such as early stopping and stronger pruning (Chu et al., 2020a; Liang et al., 2019; Wang et al., 2021b).

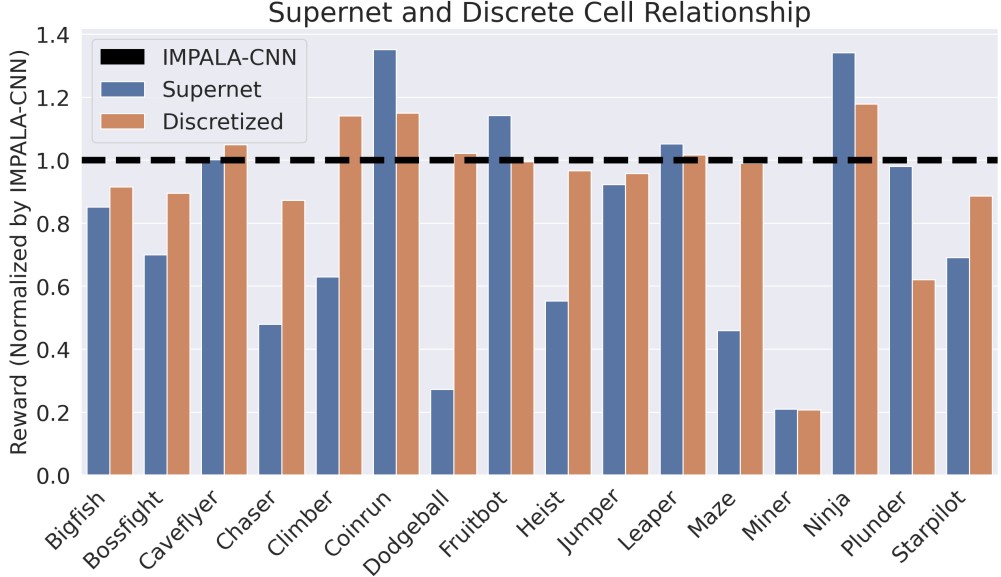

**Figure 17:** Supernet and their corresponding discrete cell rewards across all environments in Procgen using Rainbow, after normalizing using IMPALA-CNN's performances. Thus, the black dashed line at 1.0 corresponds to IMPALA-CNN.

# D   NUMERICAL SCORES

In Tables 8 and 5b, we display the average normalized reward after 25M steps, as standard in Procgen (Cobbe et al., 2020), for a subset of environments in which RL-DARTS performs competitively. The normalized reward for each environment is computed as $R_{norm} = (R - R_{min})/(R_{max} - R_{min})$ where $R_{max}$ and $R_{min}$ are calculated using a combination of theoretical maximums and PPO-trained agents, and can be found in (Cobbe et al., 2020).

| Env | IMPALA-CNN Baseline | RL-DARTS (Discrete) | Random Search |
|---|---|---|---|
| Bigfish | **0.60** | **0.60** | 0.42 |
| Bossfight | 0.75 | 0.73 | **0.81** |
| Caveflyer | 0.75 | 0.47 | **0.85** |
| Chaser | **0.71** | 0.55 | 0.15 |
| Climber | 0.69 | **0.90** | 0.61 |
| Coinrun | **0.91** | 0.53 | 0.8 |
| Dodgeball | 0.53 | **0.59** | 0.29 |
| Fruitbot | **0.92** | **0.93** | 0.83 |
| Heist | 0.72 | **0.89** | 0.38 |
| Jumper | 0.62 | 0.76 | **1.0** |
| Leaper | 0.2 | **0.28** | -0.28 |
| Maze | **1.0** | **1.0** | 0.0 |
| Miner | 0.74 | **0.85** | 0.70 |
| Ninja | **0.87** | 0.69 | 0.38 |
| Plunder | 0.57 | **0.76** | 0.43 |
| Starpilot | 0.71 | **0.73** | 0.40 |

(a) PPO + Classic

| Env | IMPALA-CNN Baseline | RL-DARTS (Discrete) | Random Search |
|---|---|---|---|
| Bigfish | **0.71** | 0.65 | 0.60 |
| Bossfight | **0.54** | **0.48** | 0.45 |
| Caveflyer | -0.05 | **-0.03** | **-0.01** |
| Chaser | **0.30** | **0.26** | 0.22 |
| Climber | **-0.04** | **-0.02** | **-0.05** |
| Coinrun | 0.06 | **0.21** | 0.15 |
| Dodgeball | **0.57** | **0.59** | -0.05 |
| Fruitbot | **0.68** | **0.68** | **0.70** |
| Heist | **-0.48** | **-0.49** | **-0.47** |
| Jumper | **0.21** | **0.18** | 0.17 |
| Leaper | -0.07 | -0.07 | **-0.03** |
| Maze | **0.76** | **0.74** | 0.51 |
| Miner | **0.35** | -0.03 | 0.11 |
| Ninja | 0.03 | 0.13 | **0.28** |
| Plunder | **0.14** | 0.02 | 0.03 |
| Starpilot | **0.91** | 0.80 | 0.76 |

(b) Rainbow + Micro

**Table 5:** Normalized Rewards in ProcGen across different search methods, evaluated at 25M steps with depths $64 \times 3$. Largest scores on the specific environment (as well as values within 0.03 of the largest) are **bolded**.

## E    EXTENDED LARGE SCALE STUDY ON RANDOM CELLS

For some environments such as Dodgeball and Maze, the gap between DARTS and random search can be dramatic. For each game, we trained 100 unique random cells using Rainbow + the "Micro" search space with depths $64 \times 3$ and found that **all of the random cells underperformed against DARTS** in Figure 18.

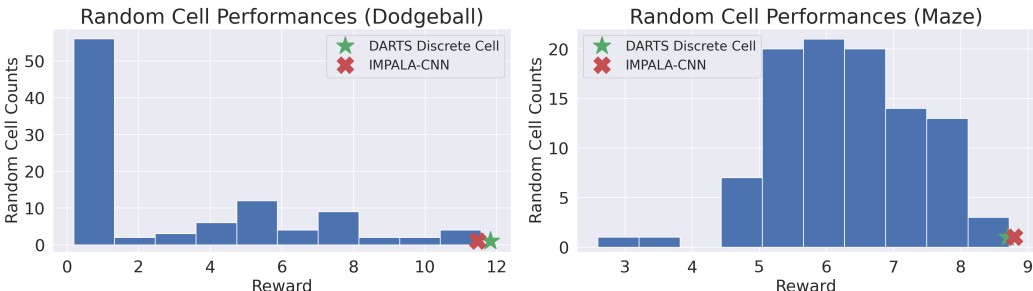

**Figure 18:** Histogram of the normalized rewards of 100 random cells evaluated on Dodgeball and Maze, using the "Micro" search space.

## F    HYPERPARAMETERS

In our code, we entirely use Tensorflow 2 for auto-differentation, as well as the April 2020 version of Procgen. For compute, we either used P100 or V100 GPUs based on convenience and availability. Below are the hyperparameter settings for specific methods. For all training curves, we use the common standard for reporting in RL (i.e. plotting mean and standard deviation across 3 seeds).

### F.1    DARTS

Initially, we swept the softmax temperature in order to find a stable default value that could be used for all environments. For PPO, the sweep was across the set $\{5.0, 10.0, 15.0\}$. For Rainbow, the sweep was across $\{10.0, 20.0, 50.0\}$.

For tabular reported scores in Figures 8 and 5b, we used a consistent softmax temperature of 5.0 for PPO, and 10.0 for Rainbow.

### F.2    RAINBOW-DQN

We use Acme (Hoffman et al., 2020) for our code infrastructure. We use a learning rate $5 \times 10^{-5}$, batch size 256, n-step size 7, discount factor 0.99. For the priority replay buffer (Schaul et al., 2016), we use priority exponent 0.8, importance sampling exponent 0.2, replay buffer capacity 500K. For particular environments (Bigfish, Bossfight, Chaser, Dodgeball, Miner, Plunder, Starpilot), we use n-step size 2 and replay buffer capacity 10K. For C51 (Bellemare et al., 2017), we use 51 atoms, with $v_{min} = 0, v_{max} = 1.0$. As a preprocessing step, we normalize the environment rewards by dividing the raw rewards by the max possible rewards reported in (Cobbe et al., 2020).

### F.3    PPO

We use TF-Agents (Guadarrama et al., 2018) for our code infrastructure, along with equivalent PPO hyperparameters found from (Cobbe et al., 2020). Due to necessary changes in minibatch size when applying DARTS modules or networks with higher GPU memory usage, we thus swept learning rate across $\{1 \times 10^{-4}, 2.5 \times 10^{-4}, 5 \times 10^{-4}\}$ and number of epochs across $\{1, 2, 3\}$.

For all models, we use a maximum power of 2 minibatch size before encountering GPU out-of-memory issues on a standard 16 GB GPU. Thus, for a $16 \times 3 = [16, 16, 16]$ DARTS supernet, we set the minibatch size to be 256, which is also used for evaluation with a $64 \times 3 = [64, 64, 64]$ discretized CNN. Our hyperparameter gridsearch for the evaluation led to an optimal setting of learning rate $= 1 \times 10^{-4}$ and number of epochs $= 1$.

# G  MISCELLANEOUS

## G.1  SEARCH SPACE SIZE

Let $O_{nz} = |\mathcal{O}| - 1$, the number of non-zero ops in $\mathcal{O}$. For a cell (normal or reduction), the first intermediate node can only be connected to the input via a single op, and thus the choice is only $O_{nz}$. However, later intermediate nodes use $K = 2$ inputs which leads to a choice size of $O_{nz}^K \times \binom{i}{K}$ where $i$ is the index of the intermediate node. Thus the total number of possible discrete cells is $O_{nz} \cdot \prod_{i=2}^{I} \left( O_{nz}^K \times \binom{i}{K} \right)$.

For the "Classic" search space, there are both normal and reduction cells to be optimized, with number of non-zero normal ops $O_{nz,N} = 5$ and number of non-zero reduction ops $O_{nz,R} = 4$, with $I = 4, K = 2$ for both. This leads to a total configuration size of $\left[ O_{nz,N} \cdot \prod_{i=2}^{I} \left( O_{nz}^2 \times \binom{i}{2} \right) \right] \times \left[ O_{nz,R} \cdot \prod_{i=2}^{I} \left( O_{nz,R}^2 \times \binom{i}{2} \right) \right] \approx 4 \times 10^{11}$.

