# OpenReview forum: "RL-DARTS: Differentiable Architecture Search for Reinforcement Learning"
_ICLR.cc/2022/Conference — ICLR 2022 Submitted_

### Official Review · Reviewer_wMrv · 2021-11-02

**Correctness:** 3
**Technical Novelty And Significance:** 2
**Empirical Novelty And Significance:** 3
**Recommendation:** 6
**Confidence:** 5

**Main Review:**

1 NAS in RL is a difficult problem. However has lot of promise. Designing suitable function approximators can affect convergence dynamics of deep RL.

2. The proposed solution and its presentation, motivation and explanation is very lucid and clear. Notations are introduced succintly and resepctive intuitions are clearly stated as well.

3. The experiments are indeed quite extensive and the authors have attempted to study the effect of apply end to end DARTS in RL from many possible dimensions including ablation studies on how the responses change in different kind of convolutions.


Weaknesses/Concerns

1. Novelty is a concern. Tts not that people have not thought about NAS for RL. but prohibitive search space becomes a bottleneck in most cases. And the authors here solve that simply by taking DARTS and applying it to learning function approximators for deep RL.

   1a.  One clear example of this is the fact that authors point out some interesting challenges wil RL earlier in the paper such as "RL training curves possess considerably more variance and noise than SL curves". However I do not see any change in formulation in section 3.2 to address that concern.

   1b. Another challenge with RL which the authors also have hinted at is that unlike SL feedback data in RL replay buffer suffers from slight distributional shift over time, until enough amount of exploration is done. This is quite a big problem and RL-DARTS formulation does take care of this as well.

 Inshort is is difficult to identify how the authors have extended DARTS formulation to suitably adapt to RL scenario. JUst stating that it was integrated with minimal code change is not sufficient.

2. The main section needs to be expanded a bit more, the entire methodology is compressed into a few paragraphs. The formal algorithm provided here is also a surprise. It literally has 3 steps. A reader can already see these 3 things. An algorithm is meant to give the reader an idea of how to even implent the same without using any backbone codebase. For instance, this is an iterative setup. So the algo should outline at each itheration, how the components of the loss are computed. in what order the architecture parameters are updated etc...

3. Diversity in domains is needed. Procgen is a great becnmark indeed. However no matter how we look at them and their different settings they are games. Other tasks such as robot locomition (simple version), or drone simulations, or treatment planning in clinical decision support systems will actually be able to realy test the validity of the claims made herein.


**Summary Of The Paper:**

The authors aim to automatically design Reinforcement Learning Q function approximator via NAS, specifically by adapting DARTS framework. They main motivation behind this is that as RL state spaces become more complex simple function approximator models, even predesigned DNNs prove to be sub-optimal, since supervised models are not always suitably designed to learn in an online fashion from reward signals rather than labels. Beyond that there are also existing RL bottlenecks such as reward sparsity, sample efficiency etc. The authors use the whole DARTS supernet at the function approximator. To bypass the bilevel optimization target in DARTS, which could prove to be intractable for RL context, authors propose to optimize the loss on cumulative returns of the RL problem by optimizing the loss on architecture and weight parameters from the replay buffer samples.

**Summary Of The Review:**

This indeed and interesting work / study and has a lot of merit. However Authors are strongly encouraged to address the concerns.

---

> ### Author Response · Authors · 2021-11-21
> **Response to Reviewer wMrv**
>
> Thank you for your positive comments and recommendation! We are glad that you found our paper’s motivation clear and our experiments comprehensive. Below, we would like to answer relevant questions and provide clarifications.
>
> ### (1) Novelty:
> “However I do not see any change in formulation in section 3.2 to address that concern [in variance and noise]”
> * To clarify, what we mean is that the **noise only affects potential blackbox optimization methods, but wouldn't affect DARTS** since DARTS uses exactly the same amount of information + data for training as the weights, as they are both using gradients with respect to the same replay buffer. More formally, the variances of the gradients are the same, i.e. $Var_{\tau}[\nabla_{\alpha} J(\pi_{\theta, \alpha})] = Var_{\tau}[\nabla_{\theta} J(\pi_{\theta, \alpha})]$. Since we know that $\theta$ trains properly using such gradients, we expect to see the same with $\alpha$. Thus DARTS is undeterred by evaluation noise and already integrates well with a standard RL pipeline.
>
> We also found some of our results surprising compared to previous literature, including:
> * Relu/Tanh “Micro search space” (for Rainbow experiments): To the best of our knowledge, there has been a lack of SL NAS work in vision for search spaces at the level of granularity of separating nonlinearities from Conv ops. Our “Micro” search space involves this granularity, and has shown to provide significant benefits (Figure 4.5) for Q-learning. Moreover, this also suggests that there may be architectures with significantly stronger inductive bias for the Q-function, prompting future research in this direction.
>
> * 16-game joint training + transferability (Section 4.5 + Figure 10): A significantly novel result we have shown is the ability to joint-train the supernet and obtain transferable cells across multiple games with very different mechanics, rules, and actions. **This suggests that RL-DARTS is even capable of operating in a very large-data multi-task regime, and thus it might be possible to find a universally strong architecture across multiple RL domains.**
>
> ## (2) Main section expansion
> Thanks for the valuable suggestion! In Appendix A, we’ve added the training process for RL-DARTS (both PPO + Rainbow), in both code-level and algorithmic details for clarity and completeness. We do note that RL-DARTS is mostly repeated replacements of the image encoder in a preexisting RL training pipeline, which demonstrates its simplicity.
>
> ## (3) Diversity of experiments:
> We are currently performing new experiments based on DM-Control [1] using another RL algorithm (Soft Actor-Critic, or SAC [2]), to show-case RL-DARTS’s versatility on **(1)** a completely different benchmark scenario in continuous control, and **(2)** a completely different training pipeline than Rainbow and PPO. Due to lack of time, we’ve updated our draft with **results in Appendix A** on selected benchmarks and we will add the full set of results in the final version of the paper.
> * From the results presented, **we find that RL-DARTS is also capable of finding better architectures than the baseline in continuous control via another optimization method**. We hope that the new analysis and results are convincing to lean even more toward acceptance of the paper.
>
> [1] Yuval Tassa et al. "DM-Control: Software and Tasks for Continuous Control" (2020)
>
> [2] Tuomas Haarnoja, Aurick Zhou, Pieter Abbeel, Sergey Levine. “Soft Actor-Critic: Off-Policy Maximum Entropy Deep Reinforcement Learning with a Stochastic Actor” (ICML, 2018)

---

> ### Author Response · Authors · 2021-11-25
> **Follow-Up to Reviewer wMrv**
>
> We thank Reviewer wMrv for their positive recommendation and suggestions for improvement. We've incorporated Points 2+3 into our draft in Appendix A with the RL-DARTS algorithm procedure as well as new positive experiments on DM-Control with the SAC algorithm. We also hope that we've clarified our novel contributions in Point 1. The discussion period is nearing an end, and we were wondering if our responses have changed some of the reviewer's opinions. We are very grateful that the reviewer provided a positive recommendation, and we hope that our new results and edits may increase the reviewer's acceptance of this work. If the reviewer has anymore questions or comments about our paper, we'd be happy to answer.
>
> Thanks in advance!

---

### Official Review · Reviewer_KUPC · 2021-11-03

**Correctness:** 4
**Technical Novelty And Significance:** 2
**Empirical Novelty And Significance:** 2
**Recommendation:** 3
**Confidence:** 4

**Main Review:**

Strength: this paper discusses several main challenges of DARTS for RL. For example, the main challenge of applying DARTS to RL is that it needs to address the non-stationary data distribution arises due to the exploration process, where the data distribution depends on the network architecture used to generate the replay data. Another challenge is that it is unclear if DARTS’s default discretization procedure leads to a better discrete cell in the RL setting due to different optimization landscape in the RL case. In addition, the work also carried out extensive experiments to examine the behaviors of DARTS for RL.

Weakness. The main weakness of the paper is its lack of novelty and insufficient technical contribution. The work is mainly an application of standard DARTS to the RL setting, without additional contributions in the algorithm side. Although the unique challenges of DARTS for RL are pointed out earlier in the paper, the paper does not propose any solution to address these challenges. In contrast, experiments in later sections seem to indicate that these are indeed not the case, and standard DARTS technique developed in SL could be directly applied to the RL setting with minimal modification. For example, experimental results seem to suggest that end-to-end DARTS works well on such non-stationary data distribution even without any additional tweaks. (Likewise, the concerns about the discretization has also been shown to be not an issue here in experiments.) While it is indeed interesting to observe that existing DARTS from SL could be readily applicable to RL, it does not play a major contribution for the current paper.

In summary, this paper can be regarded as an extensive experimental investigations, which find out that standard DARTS developed in SL are actually applicable to RL. Although the findings are interesting, it could be fit better as a workshop paper.

- Minor comments
Need to give the precise definition / expression for the loss L(theta, alpha) in (2) in the RL-DARTS.


**Summary Of The Paper:**

This paper considers the problem of differentiable neural architecture search for RL applications. It applies existing DARTS to the RL setting and conducts extensive experimental studies to examine the performance and behaviors of RL-DARTS. Experimental results show that the supernet learns alternative architectures that are highly competitive against manually designed policies.

**Summary Of The Review:**

This paper can be regarded as an extensive experimental investigations, which find out that standard DARTS developed in SL are actually applicable to RL. Although the findings are interesting, it could be fit better as a workshop paper.

---

> ### Author Response · Authors · 2021-11-19
> **Response to Reviewer KUPC, Part 1**
>
> Thank you for your feedback! It is very valuable on which parts of our paper may be unclear or too subtle, and we have corrected the relevant sections in the PDF to make our contributions more explicit. Below, we answer your questions directly. Please let us know if anything else is needed or is unclear however, as we are open to suggestions.
>
> * “Need to give the precise definition / expression for the loss L(theta, alpha) in (2) in the RL-DARTS.”
>     * In Appendix A, we’ve added a more precise and extended explanation for the construction of the loss for both Rainbow and PPO.
>
> * “unique challenges of DARTS for RL are pointed out earlier in the paper, the paper does not propose any solution to address these challenges”
>     * Apologies for the confusion. We meant that since there are no previous works for DARTS in RL, one could reasonably and **conceptually** hypothesize that these are challenges from first glance, and become concerned that regular DARTS might not work as planned in RL and would require much more complex techniques and code.
>
> * “indicate that these are indeed not the case, and standard DARTS technique developed in SL could be directly applied to the RL setting with minimal modification”
>     * Our paper **experimentally** verifies that the hypothesized challenges are no longer an issue. We believe that this verification is a core part of the novelty of our paper and is a crucial step for building a solid foundation for future DARTS-related papers in the RL. Our extensive ablations also show that the RL-DARTS optimization process performs as desired and makes sense, which mitigates any concerns about it being a potentially opaque, untrustable “hack”, which we believe is important for practitioners.
>
> Thus the structure of our introduction is that **conceptually, a RL practitioner might be hesitant to consider using DARTS** due to the non-stationarity of RL (and hence maybe why there is currently a **complete lack of DARTS applications in RL**), but **our paper experimentally shows that surprisingly, these issues do not stop even vanilla DARTS.** We believe that this opens new possibilities for widespread adoption of DARTS (and thus also efficient and easy NAS) in RL.

---

> > ### Author Response · Authors · 2021-11-19
> > **Response to Reviewer KUPC, Part 2**
> >
> > * “mainly an application of standard DARTS to the RL setting, without additional contributions in the algorithm side”
> >     * This is a very reasonable point, and we’d like to discuss in detail our thoughts and decision choices on the algorithmic portion of the paper. The core design philosophy we opted to emphasize in our work is **simplicity**, as it is very important for practitioners to quickly be able to apply RL-DARTS if they were interested in NAS. Our discretization procedure, based on **only the value of $\alpha$**, is the simplest to implement and is also agnostic to the rest of the RL pipeline, including the choice of PPO or Rainbow. Similarly, **training the supernet required zero changes to the RL algorithm and minimal hyperparameter tuning.** As we have discussed above, it turns out that the simple approach works well and provides the solid and needed experimental proof of differentiable search in RL.
> >     * However, we are aware of previous literature in improving DARTS and considered multiple new techniques, which we have also **added in Appendix A. However, these new directions can be considerably more complex** when applied in an RL system, and we provide some examples and their tradeoffs below.
> >         * Recently, discretization changes such as [1] have been found to be promising, but require iterative pruning (i.e. $\delta_{1}, \ldots , \delta_{i}, \ldots$) based on multiple calculations of reward differences between $J(\pi_{\theta, \delta_{i}})$ and $J(\pi_{\theta, \delta_{i+1}}) $ of the pruned supernet, as well as finetuning the pruned supernet $\pi_{\theta, \delta_{i}}$ at every iteration $i$. These changes, in addition to the inherently noisy evaluations of $J(\cdot)$, greatly increase the complexity of the discretization procedure, but are worth exploring in future work.
> >         * Metrics during training and performance predictors can be used for early stopping such as the Hessian [2] and Jacobian Covariance [3], but such metrics run into similar issues as the RL loss, where they must be defined with respect to the replay buffer, and thus the question is raised again as to what replay buffer data should be used. We attached preliminary plots which show that even using reasonable data from a pretrained policy’s trajectories does not provide meaningful feedback from the Jacobian Covariance.
> >
> >     * Given the above complexities, our conclusion was thus that adding in more sophisticated and incremental improvements would introduce confounding factors and may distract from the core message of this paper, which was **to set up the core foundation for differentiable search in RL.** Because of the 9 page limit, we thought it best to humbly present the main solid and factual results, and write new **separate papers about new algorithmic improvements in RL-DARTS** with comprehensive results to maintain scientific integrity. This is in line with SL, where each **SL-DARTS improvement also constitutes an entire paper alone** since DARTS research is a large subfield.
> >
> >     * We would also like to mention that in terms of **novelty in overall methods**, we introduced a new (“Micro”) search space (which is much more granular than the ones used in SL NAS) which yielded better architectures than the hand-designed IMPALA-CNN, along with 16-game joint training (whereas SL NAS only focuses on single benchmarks) which led to transferrable architectures in Section 4.5. This search space also leads to better performance on DM-Control via SAC, which we've also added in Appendix A in response to Reviewer wMrv's request.
> >
> > **Please let us know if you have any more questions/want to discuss further on this topic!** We hope that our careful analysis will convince you of the contribution’s value and change your overall appreciation of the paper.
> >
> > **References**
> >
> > [1] Ruochen Wang, Minhao Cheng, Xiangning Chen, Xiaocheng Tang, and Cho-Jui Hsieh. “Rethinking Architecture Selection in Differentiable NAS”
> >
> > [2] Arber Zela, Thomas Elsken, Tonmoy Saikia, Yassine Marrakchi, Thomas Brox, and Frank Hutter.  “Understanding and Robustifying Differentiable Architecture Search”
> >
> > [3] Colin White, Arber Zela, Binxin Ru, Yang Liu, and Frank Hutter. “How Powerful are Performance Predictors in Neural Architecture Search?”

---

> ### Author Response · Authors · 2021-11-25
> **Follow-Up with Reviewer KUPC**
>
> We thank Reviewer KUPC again for the insightful comments. Since the discussion phase is coming to an end soon, we are writing to kindly ask if the reviewer has any additional comments regarding our response. We are at their disposal for any further questions. In addition, if our clarifications and paper edits address the reviewer's concern, we would like to kindly ask if the reviewer could reconsider their score. Thanks in advance!

---

### Official Review · Reviewer_mu9V · 2021-11-03

**Correctness:** 3
**Technical Novelty And Significance:** 4
**Empirical Novelty And Significance:** 3
**Recommendation:** 6
**Confidence:** 3

**Main Review:**

The authors give a nice overview of common issues of using DARTS within the RL framework, specifically compared to supervised learning settings. An interesting point is the non-stationarity of the collected data, which depends on the initialization, which does not occur in SL.

I found the explanation of the training procedure a bit unclear, and it would be beneficial to focus on this more, as this is one of the main contributions.

The performance does not seem to be better compared to the Vanilla benchmark in 4.2. I would like to see more discussion why this happens and maybe what are some potential improvements to supernet training and discretisation that could improve the overall performance.

Some relevant references which I think should be discussed, in order to provide better context for the proposed approach:
  - Luo, Renqian, Fei Tian, Tao Qin, Enhong Chen, and Tie-Yan Liu. "Neural architecture optimization." arXiv, 2018.
  - Stanley, Kenneth O., David B. D'Ambrosio, and Jason Gauci. "A hypercube-based encoding for evolving large-scale neural networks." Artificial life, 2009


**Summary Of The Paper:**

This paper studies the use of DARTS within the RL setting. Specifically, it investigates how DARTS can optimize perception modules for RL environments. RL-DARTS is optimised in a typical RL setting, using the standard RL loss function, in an end-to-end manner.
The proposed approach is evaluated using the Procgen benchmark, and compared to the IMPALA-CNN architecture. Several ablation studies are performed to further examine the contributions of the components of the proposed approach.


**Summary Of The Review:**

- Discuss additional references
- Provide some additional discussion how can the performance be improved

---

> ### Author Response · Authors · 2021-11-19
> **Response to Reviewer mu9V**
>
> Thank you for your positive rating, suggestions on clarity, and provided references! Below, we provide direct responses and clarifications.
>
> * “explanation of the training procedure a bit unclear…”
>     * This is a very valuable comment - in Appendix A, we’ve added a more detailed explanation, which will hopefully clear up any confusions, and also shows the simplicity of our method, as it only involves replacing the image encoder in the training pipeline for both PPO and Rainbow.
>
> * “compared to the Vanilla benchmark in 4.2.”
>     * Section 4.2 only deals with the supernet training performances (which have already achieved sufficient reward); the main performance to examine is the final evaluation performance for the discrete cell. We’ve added this note in Figures 3 + 4 in Section 4.2 to clarify this point, apologies for any confusions!
> On the results in Sections 4.3-4.5 (where the proper comparisons between discrete cells and IMPALA-CNN occur), we have found that while IMPALA-CNN is a strong hand-designed baseline, there are multiple cases where RL-DARTS significantly outperforms IMPALA-CNN.
>
> * “some potential improvements to supernet training and discretisation that could improve the overall performance”
>     * We’ve added a subsection in Appendix A containing possible methods to improve RL-DARTS overall, which are definitely worth exploring in future papers.
>
> * "references which I think should be discussed":
>     * Thanks for the references! We’ve added HyperNEAT (Stanley, 2019) to the related works section and NBO (Luo, 2018) to performance-prediction inspired methods to improve RL-DARTS.

---

> ### Author Response · Authors · 2021-11-25
> **Follow-Up to Reviewer mu9V**
>
> We thank Reviewer mu9V again for their positive recommendation and suggestions. Since the discussion phase is about to end, we are writing to kindly ask if the reviewer has any additional comments or opinions regarding our response, which we hope has clarified the paper more, especially in regards to explaining the training procedure, as well as supernet performances in Section 4.2. We are very grateful for the reviewer's positive score, and hope that our new updates contribute to a stronger acceptance of the paper.
>
> Thanks in advance!

---

### Official Review · Reviewer_wzZ5 · 2021-11-05

**Correctness:** 3
**Technical Novelty And Significance:** 2
**Empirical Novelty And Significance:** 2
**Recommendation:** 5
**Confidence:** 4

**Main Review:**

[Strength]

This paper aims to tackle interesting but less explored questions: while there have been extensive studies on searching network architectures for supervised learning, it is generally less understood whether and to which extent architecture search would benefit reinforcement learning pipelines.

This paper has provided extensive experimental results exploring this direction and studied how different design choices would affect the learning procedure and final results.


[Weakness]

While I like the direction this paper is going, I have concerns regarding the novelty of this paper and whether the experiments show the full power of differentiable neural architecture search (DARTS).

The proposed method is a direct application of neural architecture search on reinforcement learning problems without any new technical innovations. I'm not against straightforward A+B types of research (in this paper, A=RL, B=DARTS). For example, the nice thing about the proposed method is that it is simple and can be a plug-and-play module on top of existing RL frameworks. However, I typically expect the combined method to accomplish tasks that neither A nor B can do, or at least demonstrate significant performance gain over existing methods. Yet from Figure 3, it seems that the proposed method is on par (in Maze) or worse (in Bigfish and Dodgeball) than the IMPALA-CNN baseline [1], which makes me worry about the significance of the proposed method and its impact on the field. For example, should we prefer IMPALA-CNN over this paper's method since IMPALA-CNN is supposed to be easier to train as the network structures are fixed? The authors may want to include more discussions or comparisons to show how the proposed method is better and in which scenario we should choose it over other state-of-the-art approaches.

Continuing from my previous point, the IMPALA-CNN baseline also seems to have a smaller search cost, according to Table 3.

One great benefit of neural architecture search is that it allows a tradeoff between model accuracy and computational efficiency [2]. Are there any similar tradeoffs we can make in the RL setting, e.g., would smaller/sparser neural networks allow higher feedback control frequency such that control becomes easier? Showing examples demonstrating the unique benefit of neural architecture search can potentially make this paper much stronger and show potentials to enable new capabilities.

In the Heist task of Figure 8, why do random search and IMPALA-CNN seem to have a very good initial start but experience a sharp drop in performance at the beginning of the training processes? Why not directly keep the initial policy?



[Minor points]

The authors made a claim in the intro that "in RL, a bad architecture ... and thus a local optimum where the loss is zero while the policy is still poor." However, the statement may not be well-grounded without context, e.g., to which objective is the "optimum" referring? Is the "loss" defined over a very small replay buffer? If the loss is zero, why is it still a local optimum (assuming that the loss is nonnegative)?

Typo in Page 7 first paragraph:
"RL time cost is partially is based on": multiple "is"


[1] Lasse Espeholt, Hubert Soyer, Remi Munos, Karen Simonyan, Vlad Mnih, Tom Ward, Yotam Doron, Vlad Firoiu, Tim Harley, Iain Dunning, Shane Legg, Koray Kavukcuoglu, "IMPALA: Scalable Distributed Deep-RL with Importance Weighted Actor-Learner Architectures"
[2] Han Cai, Ligeng Zhu, Song Han, "ProxylessNAS: Direct Neural Architecture Search on Target Task and Hardware"


===============

[Post Rebuttal]

I thank the authors for the detailed feedback. I have read the reviews from other reviewers and decided to keep my score the same (5: marginally below the acceptance threshold).

As shared by most reviewers, the novelty of this paper is a bit limited, and as I have stated in my main review, I'm not against A+B types of research (in this paper, A=Reinforcement Learning, B=Differentiable Neural Architecture Search), but I typically have high expectations from such straightforward combination.

However, I do not think this paper has shown the full benefit of the combination. It will greatly strengthen this paper if the authors show what new capabilities are enabled by the proposed method but are not possible with either A or B.

**Summary Of The Paper:**

This paper proposes to combine differentiable neural architecture search (DARTS) with standard reinforcement learning (RL) frameworks by searching the model structures (convolutional cells) for the policy and value functions. The authors have applied the method to infinitely procedurally generated Procgen benchmark and demonstrated the benefits of DARTS in RL on search efficiency in terms of time and compute. In addition, the proposed method can be easily integrated with existing RL pipelines by simply replacing the image encoder with a DARTS supernet, compatible with both off-policy and on-policy RL algorithms. The authors further show that the supernet gradually learns better cells with more training iterations, leading to alternative architectures that can be highly competitive against manually designed policies and verify previous design choices for RL policies.

**Summary Of The Review:**

While I like the direction this paper is going, I have concerns regarding the novelty of this paper and whether the experiments show the full power of differentiable neural architecture search (DARTS). Therefore I currently lean towards the rejection side. Please see my main review for more details.

---

> ### Author Response · Authors · 2021-11-19
> **Response to Reviewer wzZ5, Part 1**
>
> Thank you for the positive feedback on our paper direction! We are very grateful for your constructive comments which will improve the clarity of the paper. We’ve made edits to the paper (in blue) to improve readability and continue to improve the paper throughout the review process.
>
> Below are also direct clarifications and responses to your review, some of which can also be found in our general response to all reviewers. **Please let us know if you need anything else; we are very happy to accommodate in order to improve our score.**
>
> * "show the full power of differentiable neural architecture search (DARTS)...without any new technical innovations...more discussions or comparisons to show how the proposed method is better"
>     * This is a very reasonable point, and we've added multiple potential methods of improving RL-DARTS in Appendix A. Throughout the writing of this paper, we were aware of such methods, but decided to maintain the simplicity and core message of experimentally validating differentiable search in RL, and humbly present the main factual and empirical results in a 9-page paper, rather than potentially confound the work with algorithmic improvements. **The tradeoff for algorithmic improvement in many cases involves a more complex procedure, which also might make our work only specific to one RL algorithm, whereas we have shown that the simpler variant of DARTS is applicable to multiple RL algorithms (PPO, Rainbow, and now SAC [5] in Appendix A).** We are very interested in improving RL-DARTS, but believe that such improvements should be made in future separate papers, in line with the SL-DARTS community writing separate improvement papers, even when the SL optimization procedure can be considered simpler than RL optimization.
>
> * “Yet from Figure 3, it seems that the proposed method is on par (in Maze) or worse (in Bigfish and Dodgeball) than the IMPALA-CNN baseline”.
>     * For Figures 3 + 4 in Section 4.2, we’re describing only the supernet performance, which is expected in most DARTS scenarios to be below discrete cell (or baseline) performance. The purpose of these figures in Figure 3 was to explicitly show that supernets can train to a sufficient reward, using IMPALA-CNN as a rough gauge of what the training curve should look like. We’ve made the Figure 3+4’s captions display this purpose more.
>
>     * In terms of the **correct comparisons against IMPALA-CNN (with discretized cell evaluations), Sections 4.4 and 4.5 point out a variety of cases in which RL-DARTS outperforms IMPALA-CNN**, which is already a very strong hand-designed baseline. However, the goal of this work is not necessarily aim to beat IMPALA-CNN, but rather to provide a foundation for differentiable search techniques and their promise in RL.  In **Appendix A, we also show that RL-DARTS (via Soft Actor-Critic [5]) outperforms a default 4-layer baseline**, in response to Reviewer wMrv's request.
>
> * “IMPALA-CNN baseline also seems to have a smaller search cost, according to Table 3.”
>     * Apologies for the confusion! IMPALA-CNN is a hand-designed architecture [1], which means it doesn’t have a search cost (but its training time is used to gauge the efficiency of RL-DARTS search/supernet training, and for budgeting random search). We’ve changed the wording in Table 3.
>
> * “would smaller/sparser neural networks allow higher feedback control frequency such that control becomes easier”
>     * This is an excellent suggestion and we thank the reviewer for providing us with reference [2]. From looking at Eq. (7) in [2], **we can indeed apply similar techniques**, by adding on the differentiable latency term to the RL loss. This would be very useful in the context of large-scale robotics, where inference speed is extremely crucial for real-world deployment. We believe that our RL-DARTS paper is a foundational step in establishing the applicability of more differentiable search techniques (such as the mentioned latency reduction) in general RL pipelines, which would hopefully benefit fields such as robotics. We’ve added this reference and discussed it in our introduction + conclusion.

---

> > ### Author Response · Authors · 2021-11-19
> > **Response to Reviewer wzZ5, Part 2**
> >
> > * Heist score drop (Figure 8):
> >     * Good question. This is more of a quirk in the specific Procgen game, where initially, the policy starts with random actions, some of which can get good rewards and due to random evaluation noise, might initially log a high score on the training curve by accident. This quirk can also be found in Appendix I of the original Procgen paper [3]. However, the goal of training on this benchmark is to find a more deterministic policy which intelligently learns how to consistently maximize the reward, better than a random policy. We will make this figure cleaner by using more episodes for evaluation.
> >
> > * “to which objective is the "optimum" referring? Is the "loss" defined over a very small replay buffer? If the loss is zero, why is it still a local optimum (assuming that the loss is nonnegative)”
> >     * As an illustrative example, consider a sparse reward setting (ex: Atari’s Montezuma’s Revenge [4]), in which a poor policy will collect rollouts that result in zero rewards. In this case, for Q-learning, a trivial Q-network (which always outputs zero) has zero Bellman error loss. Similarly, for actor-critic algorithms like PPO, the advantage (reward - value function) will be zero, and thus the policy gradient becomes zero. **In both cases, the losses are zero, but the policy is still poor and cannot train due to a zeroed gradient, and thus is stuck at a “local optimum”.** This is in stark contrast to SL, where the loss is very correlated with accuracy (in classification) or is the end metric itself (in generative modelling).
> >
> >     * In RL-DARTS, we’re training supernets, a broad class of architectures that have never been used previously in regular RL. Since DARTS searches for architectures which optimize the loss rather than the reward (via training $\alpha$), hypothetically this may also lead to finding architectures which also quickly achieve 0.0 loss, but still obtain poor reward. **We refute this scenario with our experimental results, and believe that this is an important insight itself**, as it suggests RL practitioners can now simply train supernets in their own pipelines.
> >
> > * Typo: Thanks! Corrected in updated draft version.
> >
> > [1] Lasse Espeholt, Hubert Soyer, Remi Munos, Karen Simonyan, Vlad Mnih, Tom Ward, Yotam Doron, Vlad Firoiu, Tim Harley, Iain Dunning, Shane Legg, Koray Kavukcuoglu. "IMPALA: Scalable Distributed Deep-RL with Importance Weighted Actor-Learner Architectures"
> >
> > [2] Han Cai, Ligeng Zhu, Song Han. "ProxylessNAS: Direct Neural Architecture Search on Target Task and Hardware"
> >
> > [3] Karl Cobbe, Christopher Hesse, Jacob Hilton, John Schulman. “Leveraging Procedural Generation to Benchmark Reinforcement Learning”
> >
> > [4] https://paperswithcode.com/task/montezumas-revenge
> >
> > [5] Tuomas Haarnoja, Aurick Zhou, Pieter Abbeel, Sergey Levine. “Soft Actor-Critic: Off-Policy Maximum Entropy Deep Reinforcement Learning with a Stochastic Actor” (ICML, 2018)

---

> ### Author Response · Authors · 2021-11-25
> **Follow-Up to Reviewer wzZ5**
>
> We thank Reviewer wzZ5 again for the detailed review and comments, many of which we've incorporated into the updated version of the draft. Since the discussion phase is coming close to an end, we are writing to kindly ask if the reviewer has any additional comments regarding our response. We would be happy to answer any more questions about our work. In addition, if our clarifications address the reviewer's concern, we would like to kindly ask if the reviewer could reconsider their score.
>
> Thanks for your time!

---

### Author Response · Authors · 2021-11-19
**Common Response for All Reviewers, Part 1**

Thank you all for the feedback! We are very glad that all reviewers enjoyed our extensive studies and experimental verifications for DARTS in RL, which we believe will make RL-DARTS a practical and easy-to-use method for NAS in RL, one of the main goals for this paper. We’ve incorporated a variety of changes (colored in **blue** for visibility) in the updated version of the draft, based on reviewers’ comments and requests.

For the convenience of all reviewers, we put common questions and answers here:

## IMPALA-CNN Comparisons
* **Supernet training:** Section 4.2 and Figures 3+4 discuss only the supernet’s performance, which should not be used for formal comparisons. In DARTS [1], the (extremely dense) supernet is never used for final evaluation, but only for the search phase, which can also use different settings (hyperparameters, network sizes, etc.) than the baseline. The purpose of these plots is only for a sanity check, verifying that the supernet can actually train, achieving one of the goals of this paper. We should not expect the supernet to outperform the baseline IMPALA-CNN, which is only used to form a rough gauge for what a regular training curve should look like. We updated the draft to make this distinction clearer.


* **Evaluation:** What matters in NAS (common to all previous papers, e.g. [2]) is the *evaluation performance* when the discrete cell is rigorously and fairly compared to IMPALA-CNN at a larger scale, when both networks are trained from scratch using the same exact network depth + sizes and hyperparameters. Evaluations are conducted throughout the paper when we use the words “evaluation” + “discretization”.

    * In the evaluation scenario, we note that Sections 4.4 and 4.5 present many instances where **RL-DARTS has statistically significant improvements over IMPALA-CNN (and random search), much more than some of the improvements by NAS in SL**, where sometimes methods are competing against each other for only <1% accuracy over baselines [1] and random search has been argued to be even competitive [3]. As more algorithmic improvements develop for RL-DARTS, we expect to see even more gains over IMPALA-CNN.

    * However, we would like to mention that the main purpose of our work is not necessarily to beat IMPALA-CNN which turns out to be a very strong hand-designed baseline, but rather demonstrate the strong search power of RL-DARTS. As seen in Figures 9 and 18 (Appendix E), both RL-DARTS and IMPALA-CNN outperform even 100 random cells, which is an order of magnitude larger than the random search budget claimed to be competitive w/ SL NAS methods [3].


## Novelty/Technical Improvements
* We believe that one of the core novelties of this paper is experimentally finding that DARTS works well in RL without needing any new or sophisticated modifications to pre-existing methods. Our core guiding philosophy in this work is on finding a **simple, scalable solution (DARTS) to a complex problem (NAS for RL)**, and before this paper, there was a lack of experimental verification of DARTS in any optimization loops outside of pure SL, let alone in RL.

    * We did consider more advanced techniques which could improve the performance of RL-DARTS, but we wanted to keep our work simple for the sake of clarity and ease-of-use to practitioners, **shown by its applicability in 3 separate algorithms (PPO, Rainbow, and now SAC).** We believe research in **improving RL-DARTS is better for later separate papers, analogous to the SL community having separate SL-DARTS improvement papers.** We’ve provided many potential areas of improvement in Appendix A that we hope future research can build upon, but also discussed why such methods, if included to the current paper, would be significantly more complex in RL and thus confound our results and message.

* There are some technical contributions that we have also introduced in this paper which are novel among past literature:

    * **16-game joint training + transferability (Section 4.5 + Figure 10):** A significantly novel result we have shown is the ability to joint-train the supernet and obtain transferable cells across multiple games with very different mechanics, rules, and actions. This suggests that RL-DARTS is even capable of operating in a very large-data multi-task regime, and thus it might be possible to find a universally strong architecture across multiple RL domains.

    * **Relu/Tanh “Micro search space” (for Rainbow and new SAC experiments):** To the best of our knowledge, there has been a lack of SL NAS work in vision for search spaces at the level of granularity of separating nonlinearities from Conv ops. Our “Micro” search space involves this granularity, and has shown to provide significant benefits (Figure 4.5) for Q-learning. Moreover, this also suggests that there may be architectures with significantly stronger inductive bias for the Q-function, prompting future research in this direction.

---

> ### Author Response · Authors · 2021-11-19
> **Common Response for All Reviewers, Part 2**
>
>
> ## Explaining RL-DARTS Procedure
>
> In Appendix A, we’ve added the exact description of the standard PPO + Rainbow training methods when using a variable image encoder, and thus RL-DARTS can simply be restated as swapping in/out a supernet or discretized network for this image encoder. We hope that this adds more clarity but also shows the incremental simplicity of adding DARTS into RL.
>
> **References:**
>
> [1] Hanxiao Liu, Karen Simonyan, Yiming Yang. “DARTS: Differentiable Architecture Search”
>
> [2] Barret Zoph, Vijay Vasudevan, Jonathon Shlens, Quoc V. Le. “Learning Transferable Architectures for Scalable Image Recognition”
>
> [3] Liam Li, Ameet Talwalkar. “Random Search and Reproducibility for Neural Architecture Search”

---

### Decision · Program_Chairs · 2022-01-20

**Decision:**

Reject

**Comment:**

The paper studies network architecture search in the context of reinforcement learning. In particular it applies the DARTS method to the Procgen RL benchmark, and conducts extensive experimental evaluations. It identifies a number of issues that could potentially prevent DARTS from working well in the RL setting (such as nonstationarity and high variance), but in the end shows good performance without needing to modify DARTS substantially.

The reviewers agreed that a key strength of the paper is in its experiments. But they also identified a weakness in novelty: if a paper's main contribution is to combine two previously well-explored ideas (in this case, RL and DARTS) then there is a high bar for the quality of exposition and positioning, and the reviewers did not feel that this bar was met. (Though the authors' updates during the rebuttal period did help substantially with clarity and relationship to other methods -- thank you for these!)

Recommended decision: while the paper makes a worthwhile contribution, it does not in its current form rise to the level of novelty and general interest that is needed for publication in ICLR.